# Synthetic biodegradable microporous hydrogels for in vitro 3D culture of functional human bone cell networks

Doris Zauchner [1], Monica Zippora Müller[1], Marion Horrer [1], Leana Bissig [1], Feihu Zhao [2], Philipp Fisch [1], Sung Sik Lee [3], Marcy Zenobi-Wong [1], Ralph Müller [1] & Xiao-Hua Qin [1] ✉

Generating 3D bone cell networks in vitro that mimic the dynamic process during early bone formation remains challenging. Here, we report a synthetic biodegradable microporous hydrogel for efficient formation of 3D networks from human primary cells, analysis of cell-secreted extracellular matrix (ECM) and microfluidic integration. Using polymerization-induced phase separation, we demonstrate dynamic in situ formation of microporosity (5–20 μm) within matrix metalloproteinase-degradable polyethylene glycol hydrogels in the presence of living cells. Pore formation is triggered by thiol-Michael-addition crosslinking of a viscous precursor solution supplemented with hyaluronic acid and dextran. The resulting microporous architecture can be fine-tuned by adjusting the concentration and molecular weight of dextran. After encapsulation in microporous hydrogels, human mesenchymal stromal cells and osteoblasts spread rapidly and form 3D networks within 24 hours. We demonstrate that matrix degradability controls cell-matrix remodeling, osteogenic differentiation, and deposition of ECM proteins such as collagen. Finally, we report microfluidic integration and proof-of-concept osteogenic differentiation of 3D cell networks under perfusion on chip. Altogether, this work introduces a synthetic microporous hydrogel to efficiently differentiate 3D human bone cell networks, facilitating future in vitro studies on early bone development.

Bone development—or osteogenesis—is a complex process that involves substantial changes in cell morphologies owing to a dynamic interplay between bone cells and their extracellular matrix (ECM)[1]. At an early stage of osteogenesis, osteoblasts lay down a thin layer of collagen-rich matrix (osteoid)[2]. Thereafter, cells gradually embed themselves into this matrix to form a three-dimensional (3D) osteocyte network while inducing matrix mineralization[3]. This process generates an intricate network of fluid-filled tunnels (i.e., lacuno-canalicular system), which have proven to be crucial for mediating load-induced bone formation[4–7].

Although traditional bone tissue engineering centered on generating mineralized bone-like implants, there is a trend towards creating in vitro models of bone development for disease modeling and drug testing. Two major approaches have been sought: cell embedding in hydrogels and top seeding on porous scaffolds (Fig. 1). For 3D embedding, a variety of natural (e.g., collagen[8], gelatin[9], alginate-collagen[10]) and synthetic hydrogels (e.g., clickable polyvinyl alcohol[11] or poly(ethylene glycol) (PEG)[12]) have been reported. However, these hydrogels often have nanoscale pore sizes (5–100 nm) and

[1]Institute for Biomechanics, ETH Zurich, Zurich, Switzerland. [2]Department of Biomedical Engineering and Zienkiewicz Centre for Computational Engineering, Swansea University, Swansea, UK. [3]Institute of Biochemistry and Scientific Center of Optical and Electron Microscopy, ETH Zurich, Zurich, Switzerland. ✉e-mail: qinx@ethz.ch

**Fig. 1 | Schematic illustration of conventional biomaterials versus microporous polyethylene glycol (PEG) hydrogels for bone tissue engineering.** Traditional nanoporous hydrogels (pore sizes: 5–100 nm[13]) often impede cell spreading, while macroporous scaffolds with large pores (pore sizes: 100–600 μm[14–17]) merely provide a 2D cell surface interface. Herein, microporous PEG hydrogels (pore sizes: 5–20 μm) are developed for rapid 3D cell network formation. Illustration, created with BioRender.com, released under a Creative Commons Attribution-NonCommercial-NoDerivs 4.0 International license.

limited permeability[13]. Consequently, cell spreading relies on matrix degradation via hydrolysis or proteolysis through cell-secreted matrix metalloproteinases (MMPs). In contrast, top seeding typically relies on scaffolds with large pores (100–600 μm)[14–17] where the cell-to-surface interface is 2D. As a result, cells often fail to form 3D networks. Various techniques, such as emulsification[18], porogen leaching[19–22], and particle or microgel annealing[23–25], have been employed to create microporous hydrogels with relevant pore sizes of 5–150 μm to facilitate cell spreading. Nevertheless, most of these methods have limitations in generating interconnected pores in the presence of living cells, while others require the degradation of a sacrificial porogen phase and, therefore, multiple processing steps.

In the last decade, microfluidic organs-on-a-chip technology found increasing applications in tissue engineering[26–28]. These tools provide exquisite control over environmental cues, such as mechanical and biochemical stimuli. Efforts to create bone-on-chip models have been actively sought for. For instance, Nasello et al.[29] reported a static microfluidic culture of primary human osteoblasts in a collagen matrix. Notably, higher cell seeding density promoted osteogenic differentiation. Yet, using collagen limits the analysis of cell-secreted ECM, an important hallmark of bone development. Very recently, Bahmaee et al.[30] combined porous polymerized high internal phase emulsion with a custom chip for the perfusion culture of progenitor cells. To date, however, cells in current models fail to form a functional 3D cell network.

Herein, we present a synthetic biodegradable microporous hydrogel for in vitro generation of functional human bone cell networks in 3D and microfluidic integration. Given the crucial role of MMPs in bone development, we employ MMP-sensitive, microporous PEG hydrogels to study the role of matrix degradability in the osteogenic differentiation of human primary cells as well as ECM production. By optimizing the pore size and permeability, we demonstrate proof-of-concept integration of these hydrogels with a microfluidic chip and perfusion osteogenic culture.

## Results and discussion
### Design of microporous PEG hydrogels
In the present work, a synthetic void-forming hydrogel was designed to generate 3D bone cell networks to mimic the process of osteoblast embedding during early osteogenesis. PEG hydrogels were formed by thiol-Michael crosslinking[31] between 4-arm PEG vinylsulfone (4-PEG-VS) and di-thiol crosslinkers in the presence of dextran and hyaluronic acid (HA). Polymerization-induced phase separation (PIPS) is a

dynamic process where an initially miscible mixture (single-phase) undergoes phase decomposition during polymerization of the reactive components, thereby forming a multi-phase blend[32]. Upon mixing the components (4-PEG-VS, HA, dextran, and crosslinker) and elevating the temperature to 37 °C, the mixture undergoes PIPS induced by Michael-addition crosslinking (Fig. 2a) and forms PEG hydrogels with interconnected porosity through a single processing step[33]. HA is necessary to increase the complex viscosity in a suitable range (0.3–0.6 Pa s) and thus prevent the phases from collapsing during PIPS. A fibronectin-derived arginylglycylaspartic acid peptide (N-C: CG**RGD**SP) was added to promote cell attachment, whereas a matrix metalloproteinase (MMP)-sensitive di-cysteine peptide[11,34] (N-C: KCGPQG↓IWGQCK or GCRDGPQG↓IWGQDRCG, ↓ indicates cleavage site) and PEG di-thiol[33] (PEG-2-SH, $M_W$ = 2.0 kDa or 3.4 kDa) were used as degradable and non-degradable crosslinkers, respectively. Using fluorescently labeled 4-PEG-VS, dynamic in situ pore formation was evidenced by time-lapsed confocal microscopy. During crosslinking, spatial patterns of binodal nucleation were observed (Fig. 2b, Supplementary Movie 1). Supplementary Fig. 1 and Supplementary Movie 2 show that PIPS and interconnected porosity were only observed in the phase-separated hydrogels using dextran.

### Effect of polymer composition on hydrogel mechanics
Rheological analysis revealed that increasing the concentration of 4-PEG-VS from 2.0% to 2.5% yielded significantly stiffer hydrogels for both degradable and non-degradable groups with storage moduli ($G'$) ranging from 30 ± 17.2 Pa to 421 ± 169.5 Pa (Fig. 2c). Depending on the hydrogel composition, gelation (crossover of $G'$ and loss modulus ($G''$)) started immediately upon in situ crosslinking at 37 °C and plateaued after 60 min. MMP-degradable hydrogels crosslinked faster than their non-degradable counterparts (Supplementary Fig. 2a). Lutolf et al.[35] demonstrated that the presence of positively charged amino acid residues near the cysteine of a crosslinker, such as the lysine residue in the MMP-sensitive crosslinker, modulates the thiol group's p$K_a$, thereby accelerating crosslinking kinetics in Michael-addition PEG hydrogels.

By increasing the dextran concentration from 0.0% to 1.0%, $G'$ increased for both non-degradable and degradable hydrogels. This effect can be attributed to the strong phase-separating properties of dextran against PEG precursors, which may lead to localized densification of the PEG phase and stiffer hydrogels. Varying the dextran molecular weight ($M_W$, 40 or 500 kDa), however, did not significantly affect the hydrogel mechanical properties (Supplementary Fig. 2b). By

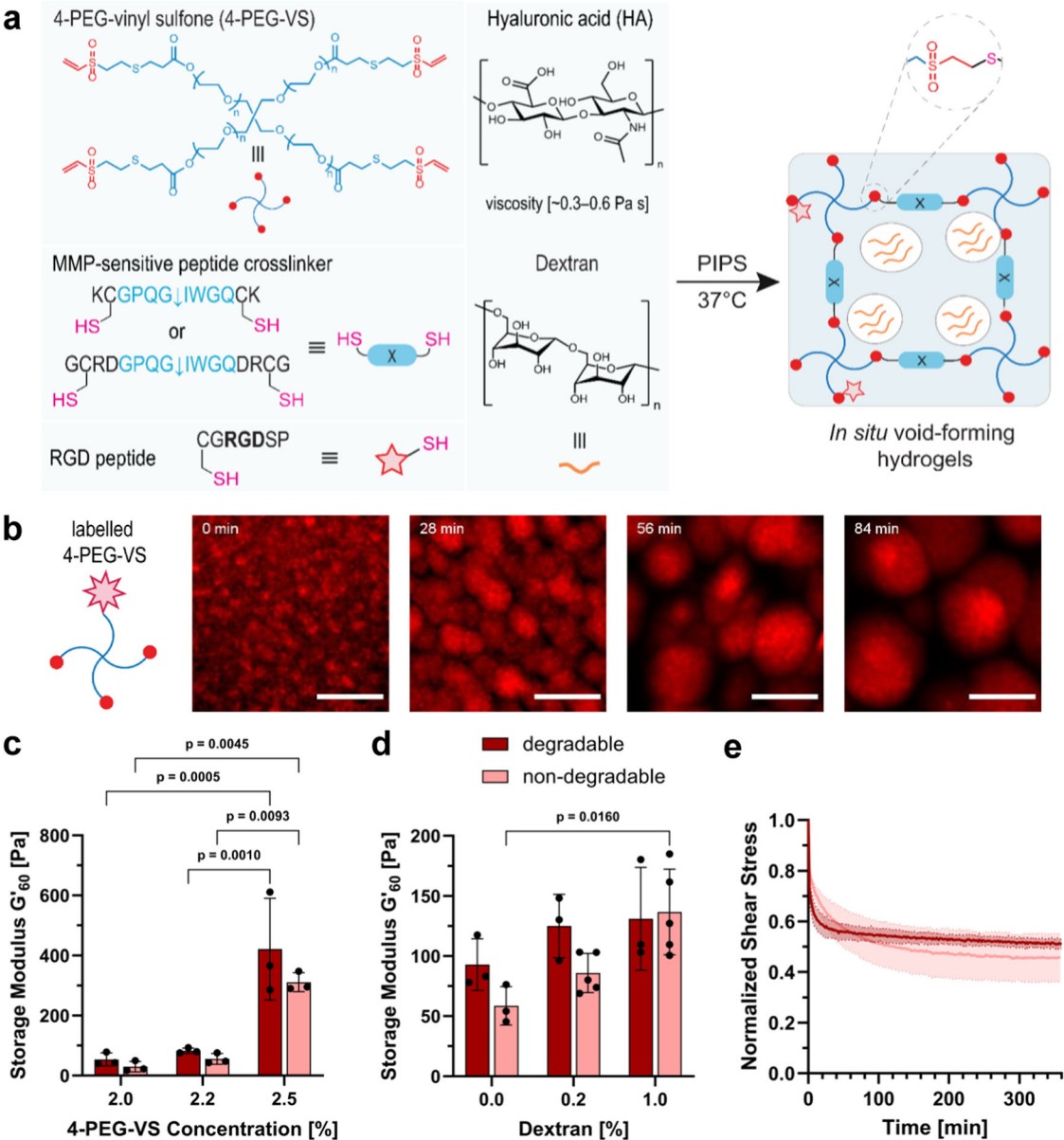

**Fig. 2 | Synthesis and characterization of microporous PEG hydrogels by polymerization-induced phase separation (PIPS). a** Illustration of in situ pore formation by PIPS at 37 °C: upon addition of a matrix metalloproteinase (MMP)-degradable di-cysteine crosslinker, 4-arm PEG vinylsulfone (4-PEG-VS) is cross-linked in the presence of dextran and hyaluronic acid (HA), leading to in situ pore formation. **b** Time-lapsed confocal microscopy images showing PIPS between rhodamine-labeled 4-PEG-VS and dextran, scale bar: 10 μm. **c** Storage modulus ($G'$) of microporous hydrogels with varying 4-PEG-VS concentration and MMP-degradable peptide or PEG di-thiol as non-degradable crosslinker after 60 min of crosslinking at 37 °C, $n = 3$ samples (mean ± SD, two-way ANOVA/Tukey). **d** $G'$ of degradable and non-degradable microporous hydrogels with varying dextran concentration after 60 min of crosslinking at 37 °C, $n = 5$ samples for non-degradable 0.2% and 1.0% dextran and $n = 3$ samples for all other groups (mean ± SD, two-way ANOVA/Tukey). **e** Stress-relaxation of PEG hydrogels measured by rheology at 5% constant strain, $n = 3$ samples (mean ± SD).

changing the concentration of HA and thereby the viscosity of the hydrogel precursor, the stiffness of the hydrogel could be tuned (Supplementary Fig. 2c). High viscosity has been suggested to prevent the phases from collapsing into microspheres before the structures are stabilized by crosslinking in PIPS[33]. Our findings show that the inclusion of HA enhanced the crosslinking when its concentration was increased from 0.25% to 0.50%. However, the higher concentration of HA (0.83%) reduced $G'$, indicating that crosslinking was less efficient for highly viscous formulations. Interestingly, a rheological test evidenced the presence of viscoelastic behavior in both degradable and non-degradable PEG hydrogel matrices, likely attributed to the presence of HA (Supplementary Fig. 2d). We have recently demonstrated

that the presence of a viscous gelatin component in synthetic PVA hydrogels provides stress-relaxation properties and enables rapid cell spreading[36]. Our rheological data shows that degradable and non-degradable PEG hydrogels containing HA exhibit stress relaxation at a constant strain of 5% (Fig. 2e). This property may facilitate cytoskeletal rearrangements and the formation of a 3D cell network.

It is important to note that rheological analysis of the bulk hydrogels may not reflect the local hydrogel mechanics experienced by cells. Due to the low $G'$, atomic force microscopy-based nanoindentation was unsuccessful. In the future, microrheology using nano-sized beads may help dissect how mechanical heterogeneity in microporous hydrogels impacts cell behaviors[37].

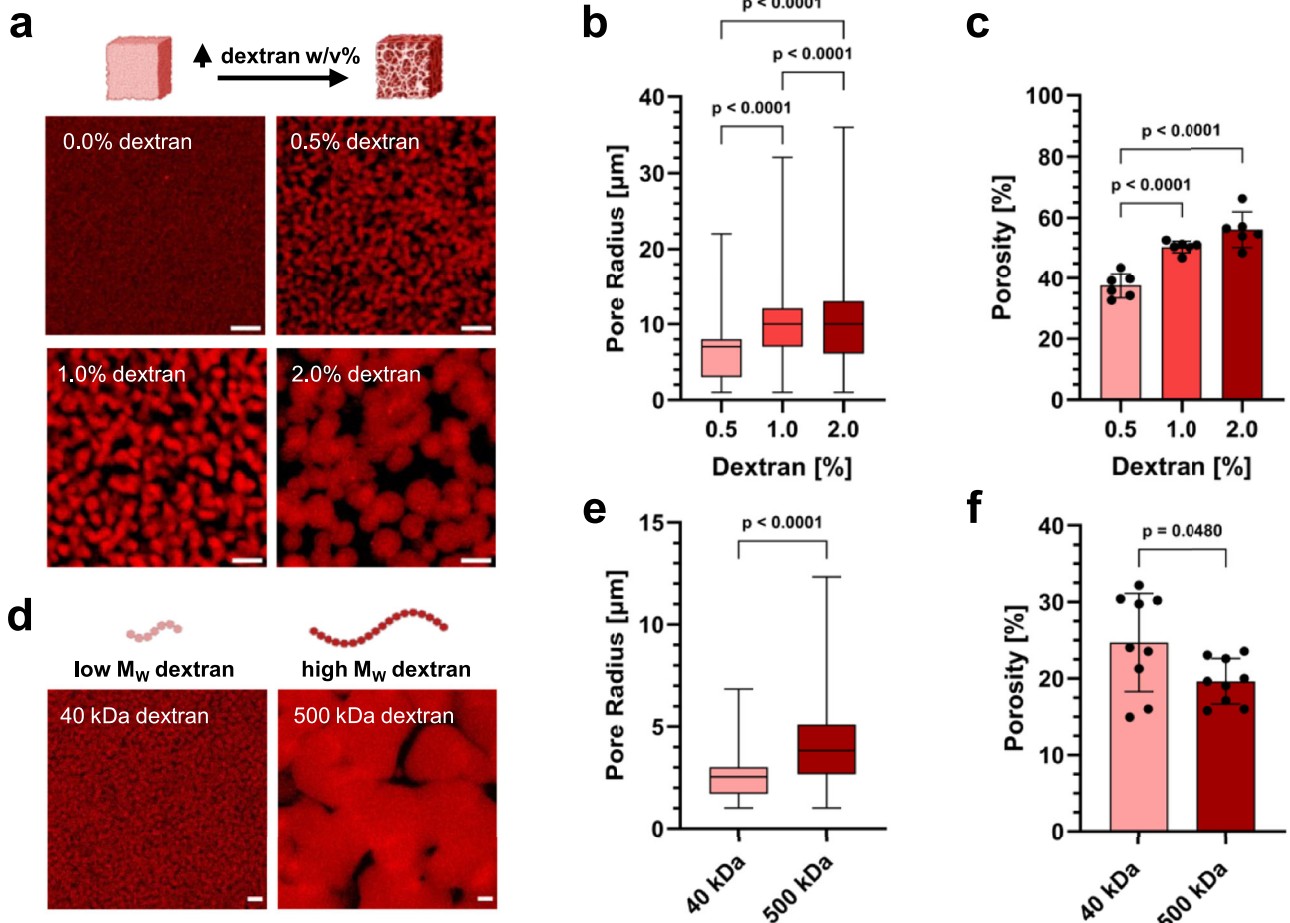

**Fig. 3 | Characterization of the porous architecture of void-forming PEG hydrogels in function of dextran concentrations and molecular weights.** **a** Confocal microscopy images of rhodamine-labeled PEG hydrogels formed with varying dextran concentrations without physical constraints, scale bars: 10 µm. **b** and **c** Quantification of pore radius and porosity of hydrogels formed with varying dextran concentrations, $n = 3$ samples with $n = 2$ imaging positions each (one-way ANOVA/Tukey). **d** Confocal microscopy images of rhodamine-labeled PEG hydrogels formed with 1.0% low $M_W$ (40 kDa) and high $M_W$ (500 kDa) dextran without physical constraints, scale bars: 10 µm. **e** and **f** Quantification of pore radius and porosity of hydrogels formed with low $M_W$ (40 kDa) and high $M_W$ (500 kDa) dextran, $n = 3$ samples with $n = 3$ imaging positions each (two-sided Student's $t$-test). Box plots in **b** and **e** show the 25th and 75th percentiles at the lower and upper limit, respectively, whiskers indicate minimum and maximum values, center line indicates the median. Bar plots in **c** and **f** show mean ± SD. Illustrations in panels **a** and **d**, created with BioRender.com, released under a Creative Commons Attribution-NonCommercial-NoDerivs 4.0 International license.

## Characterization of pore architecture

We investigated the impact of dextran inclusion on pore architecture in PEG hydrogels. The porosity after PIPS was analyzed by quantification of confocal microscopy images using an algorithm reported by Vandaele et al.[38] As shown in Fig. 3a–c and Supplementary Fig. 3, the increase of dextran concentration from 0.5% to 1% and 2% led to an increase in pore radius (median pore radius: 7.0, 10.0 and 10.0 µm, respectively) and porosity (37.6 ± 4.1, 50.4 ± 1.9 and 55.8 ± 5.9%, respectively) as well as higher pore connectivity. The pore size in hydrogels without dextran was in the sub-µm range and thus too small to be quantifiable. Furthermore, the increase of dextran $M_W$ from 40 to 500 kDa resulted in a shift towards larger pore radii (Fig. 3d, e, Supplementary Fig. 4). The resultant porosity in 500 kDa dextran group (19.7 ± 3.0%) was smaller than that of the 40 kDa group (24.7 ± 6.4%) (Fig. 3f). Furthermore, we assessed the effect of physical constraints on pore formation (Supplementary Fig. 5) inside a microfluidic channel. Similar to in non-constrained hydrogels, the pore radius for the high $M_W$ dextran (500 kDa) group was larger (median: 4.2 µm) than that of the low $M_W$ dextran (40 kDa) group (median: 2.6 µm). Moreover, pore connectivity on chip was increased with higher dextran $M_W$. These results demonstrate that the pore architecture of PEG hydrogels can be fine-tuned by dextran concentration and $M_W$. The injectability of these

synthetic hydrogels, along with their ability to retain void-forming characteristics when cast on a chip, holds promise for microfluidic cell culture applications.

## Rapid formation and long-term stability of 3D bone cell networks

In this study, we investigated whether our microporous PEG hydrogels enable in vitro generation of 3D bone cell networks. Human mesenchymal stromal cells (hMSC) were embedded inside MMP-degradable and non-degradable PEG hydrogels and differentiated under osteogenic conditions. MMPs are known to be crucial for hMSC differentiation as well as for osteoblast survival during osteogenesis[39]. Therefore, we reason that only when cells are able to remodel their surrounding matrix through proteolysis by MMPs can they form an interconnected 3D cell network with long-term stability (Fig. 4a). hMSC viability on day 2 was above 80% in both degradable and non-degradable hydrogels (Fig. 4b). Notably, ultrafast cell spreading as well as 3D cell network formation within these microporous hydrogels were observed respectively as early as 1.5 and 24 h after being embedded in the hydrogel (Supplementary Fig. 6). When embedded in MMP-degradable hydrogels, hMSC continuously remodel their environment and maintain a 3D cell network for at

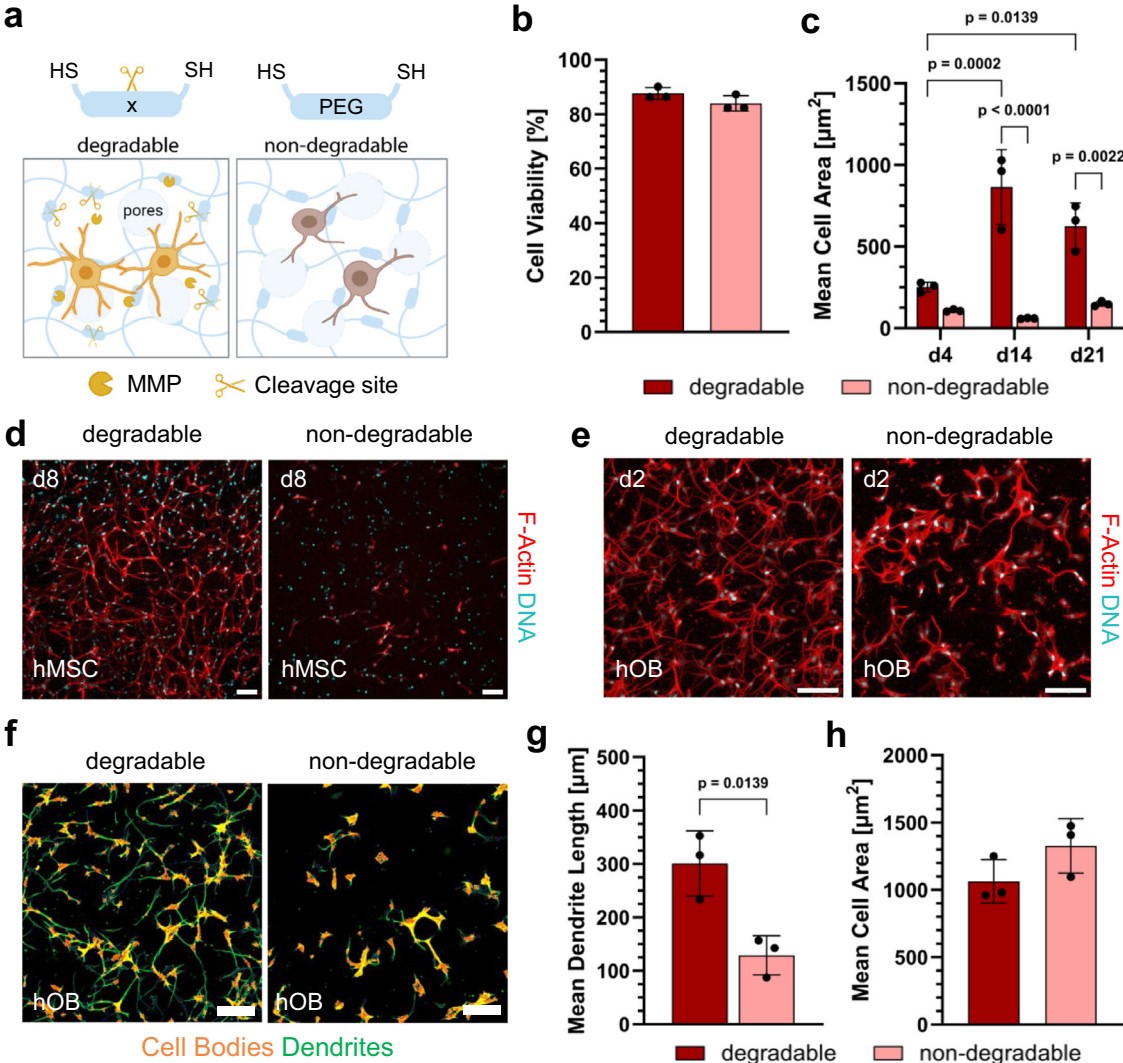

**Fig. 4 | Static human bone cell culture within degradable and non-degradable microporous PEG hydrogels. a** Illustration of the formation of 3D cell networks in degradable hydrogels (left) and the degeneration of cell networks in non-degradable hydrogels (right). Illustration, created with BioRender.com, released under a Creative Commons Attribution-NonCommercial-NoDerivs 4.0 International license. **b** The impact of hydrogel composition on human mesenchymal stromal cell (hMSC) viability on day 2, $n = 3$ samples (mean ± SD, two-sided Student's *t*-test). **c** The impact of hydrogel composition on mean hMSC area after 4, 14, and 21 days of 3D osteogenic culture, $n = 3$ samples (mean ± SD, two-way ANOVA/ Tukey). **d** Representative maximum intensity projections (MIPs) of actin-nuclei-stained hMSC networks in degradable and non-degradable hydrogels after 8 days of culture, scale bars: 50 μm. **e** Representative MIPs of actin-nuclei-stained human osteoblast (hOB) networks in degradable and non-degradable hydrogels after 2 days of culture, scale bars: 100 μm. **f** Semiautomatic labeling of hOB cell bodies and dendrites by NeuriteQuant plugin in ImageJ for quantitative comparisons of cell morphologies, scale bars: 100 μm. **g** Quantification of mean hOB dendrite length per cell using NeuriteQuant in degradable and non-degradable hydrogels, $n = 3$ samples (mean ± SD, two-sided Student's *t*-test). **h** Quantification of mean hOB cell area in degradable and non-degradable hydrogels, $n = 3$ samples (mean ± SD, two-sided Student's *t*-test).

least 35 days. Quantification of mean cell area further demonstrated the permissiveness of degradable hydrogels for hMSC (Fig. 4c). The average cell area in the degradable hydrogels was significantly larger compared to non-degradable hydrogels. An extensive 3D cell network formed in the degradable hydrogels on day 8. By contrast, the extent of cell network formation was significantly less in the non-degradable hydrogels (Fig. 4d, Supplementary Fig. 7). When grown in a more permissive environment, hMSC formed multiple dendritic processes connecting them with neighboring cells.

Based on the success of hMSC cultures, we further tested the suitability of microporous PEG hydrogels for 3D osteogenic culture of human osteoblasts (hOB). Cell viability after embedding was above 90% in both degradable and non-degradable groups and remained high after 2 days of culture (Supplementary Fig. 8). Similar to hMSC, a difference in cell morphology between degradable and non-

degradable hydrogels was observed (Fig. 4e). Quantification of cell denticity using ImageJ NeuriteQuant[40] (Fig. 4f) shows that hOB within degradable PEG gels have longer dendritic processes (Fig. 4g) despite having similar mean cell areas (Fig. 4h). Moreover, we observed that the addition of cell-adhesive RGD-peptides is critical for hOB spreading and network formation (Supplementary Fig. 9). Notably, we demonstrated that manipulating the porous architecture by increasing dextran concentration and, consequently, pore size influenced hOB morphology (Supplementary Fig. 10a). Cells within PEG hydrogels with larger pore sizes (1.0% dextran) displayed a significantly larger cell area (Supplementary Fig. 10b) and longer dendrites (Supplementary Fig. 10c) on the second day of osteogenic culture compared to hydrogels with smaller pore sizes (0.2% dextran). In summary, our findings underscore the importance of both matrix degradability and porous architecture in the formation of long-term stable 3D cell networks from both hMSC and hOB.

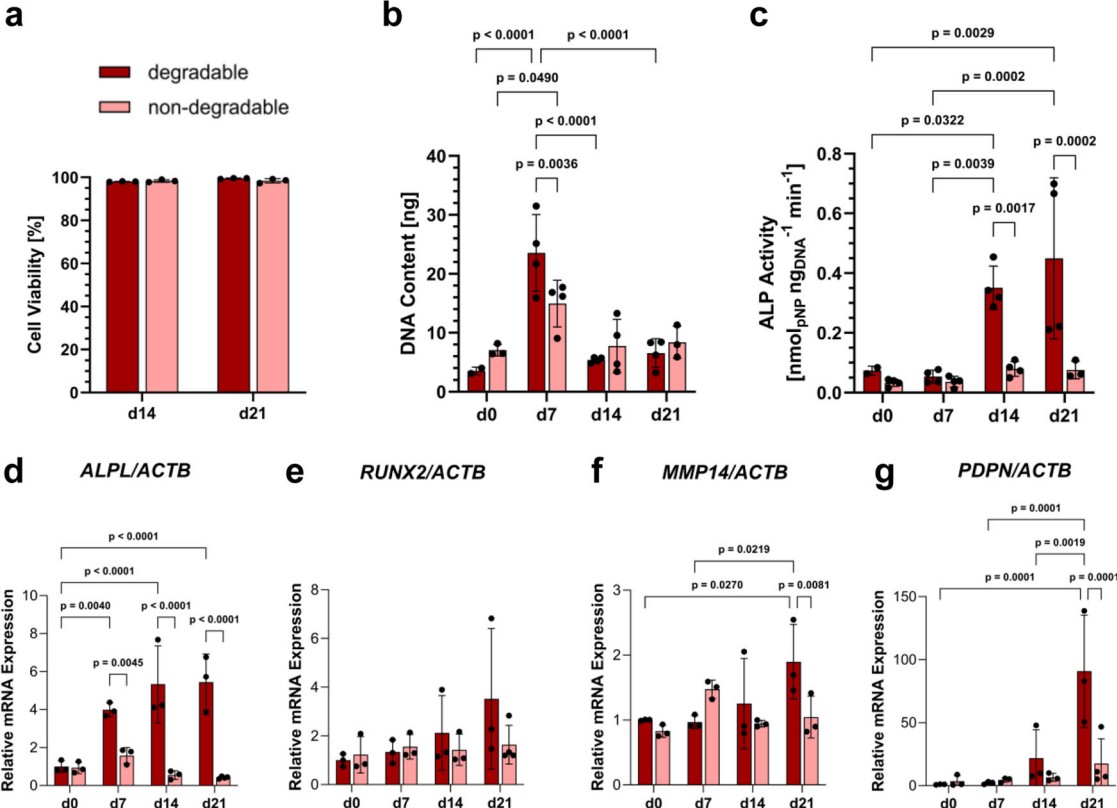

**Fig. 5 | Quantitative analysis of static 21-day osteogenic hMSC culture within MMP-degradable and non-degradable PEG hydrogels. a** Cell viability of hMSC after 14 and 21 days of osteogenic culture based on live/dead staining, $n = 3$ samples (mean ± SD, two-way ANOVA/Tukey). **b** DNA assay of 3D hMSC culture in degradable and non-degradable PEG hydrogels, $n = 2$ samples for degradable d0, $n = 3$ for non-degradable d0 and d21 and $n = 4$ for all other groups (mean ± SD, two-way ANOVA/Tukey). **c** ALP activity (measured as p-nitrophenol concentration) of 3D

hMSC culture in degradable and non-degradable PEG hydrogels normalized to DNA content, $n = 2$ samples for degradable d0, $n = 3$ for non-degradable d0 and d21 and $n = 4$ for all other groups (mean ± SD, two-way ANOVA/Tukey). **d–g** Quantification of gene expression levels using qPCR for markers of osteoblasts (*ALPL*, *RUNX2*), matrix degradation (*MMP14*), and early osteocytes (*PDPN*) over 3 weeks normalized to *ACTB* expression, $n = 4$ samples for non-degradable d21 and $n = 3$ for all other groups (mean ± SD, two-way ANOVA/Tukey).

## Osteogenic differentiation of hMSC in static PEG hydrogel cultures

Next, we investigated the osteogenic differentiation of hMSC in microporous PEG hydrogels as a function of matrix degradability. A live-dead staining assay revealed high cell viability (>95%) in both types of hydrogels at 14 and 21 days (Fig. 5a, Supplementary Fig. 11). A DNA assay indicated a significant initial increase in cell numbers within both groups over the first 7 days, followed by a reduction in cell numbers (Fig. 5b). Alkaline phosphatase (ALP) activity was significantly higher in degradable hydrogels at 14 and 21 days (Fig. 5c). Furthermore, real-time quantitative PCR (RT-qPCR) data showed a pronounced increase in *ALPL* gene expression in the degradable hydrogels from day 7 to day 21, compared to the non-degradable ones (Fig. 5d). *RUNX2* gene expression was higher in degradable hydrogels at 14 days (Fig. 5e). Additionally, *MMP14* and *PDPN*, involved in osteocytic dendrite formation[41], were expressed to a greater extent in degradable hydrogels (Fig. 5f, g). Immunofluorescence staining revealed expression of collagen type I in both hydrogels over time (Supplementary Fig. 12a, c; negative controls in Supplementary Fig. 13). However, collagen I appeared more widely distributed across the degradable hydrogels compared to the non-degradable ones. Notably, the expression of podoplanin was significantly higher in degradable PEG hydrogels than in non-degradable ones at day 21 (Supplementary Fig. 12b, d).

As the osteogenic differentiation proceeded, the cell-laden hydrogels became increasingly opaque. As such, histological sections of the same cultures on day 8 (Supplementary Fig. 14) and day 30 (Fig. 6) were prepared to further compare cell morphology, collagen

secretion, matrix mineralization, and osteocalcin expression. Similar to the non-sectioned samples, results show that degradable hydrogels facilitated pronounced cell network formation on day 8 (Supplementary Movie 3) that remained stable for at least 30 days in contrast to non-degradable hydrogels. Picrosirius red-polarization imaging revealed the presence of cell-secreted collagen fibers on day 8, especially in the MMP-degradable hydrogels, due to its permissiveness for cell-matrix remodeling. Importantly, the synthetic nature of the microporous PEG hydrogels allowed for the assessment of collagen secretion, considering that all detectable collagen fibers were produced by embedded cells. This is unachievable in conventional proteinaceous hydrogels such as collagen type I[29] and gelatin derivatives[42]. Collagen content and maturity were quantified based on the fiber hue method as described elsewhere[43]. Fig. 6b shows that more green and yellow color corresponding to low fiber thickness and immature collagen was present in the non-degradable hydrogels. In contrast, cells within the MMP-degradable hydrogels produced more mature collagen fibers, as indicated by the larger proportion of red color (Fig. 6c). Moreover, the red color content significantly increased from day 8 to day 30. Alizarin red staining further indicated enhanced matrix mineralization, especially in close proximity to embedded cells within the degradable hydrogels compared to non-degradable ones after 30 days (Fig. 6d). Compared to day 8, an increase in mineral deposition on day 30 implies the 3D osteogenic differentiation of hMSC into a more mature bone cell phenotype. Osteocalcin, a marker for mature osteoblasts[3], was predominantly expressed in MMP-degradable hydrogels after cultivation for 30 days. In contrast, only limited

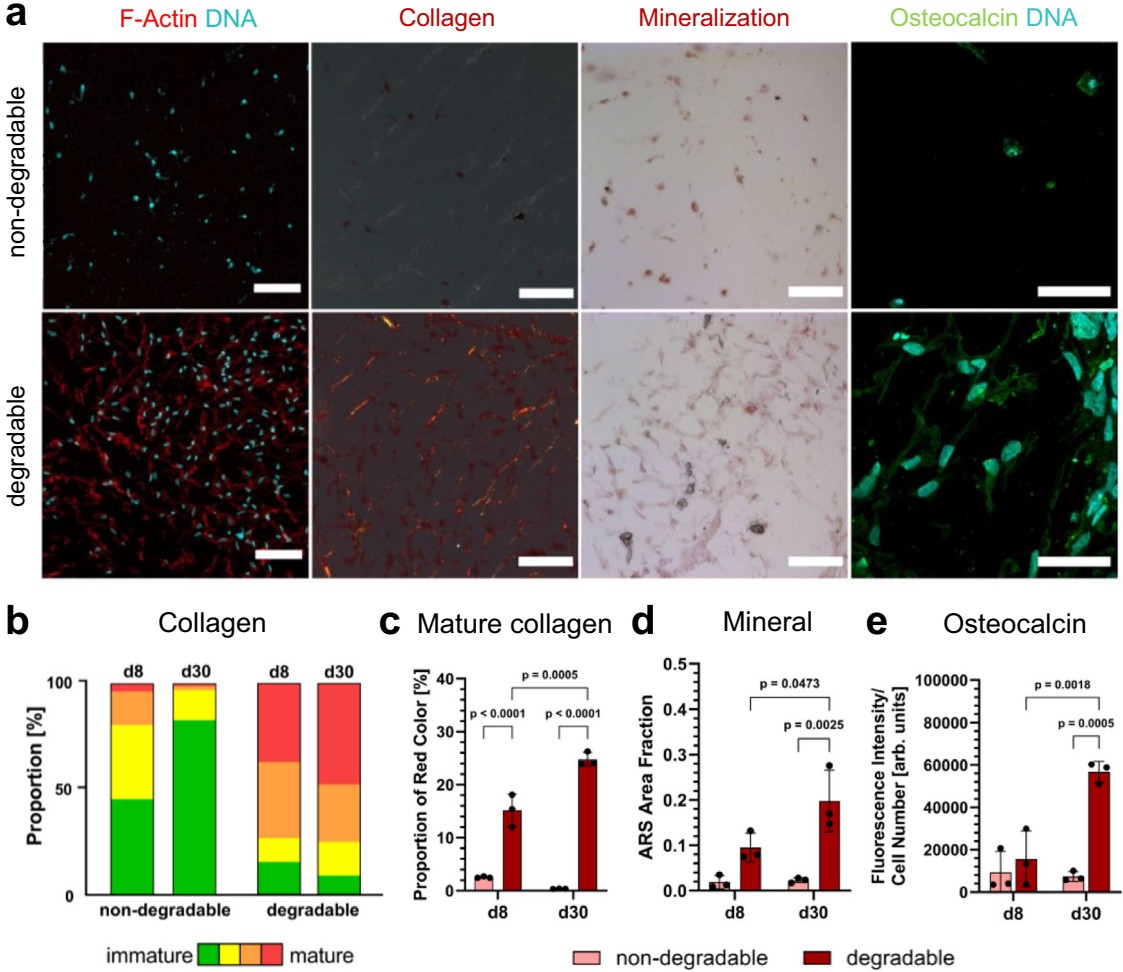

**Fig. 6 | Histological analysis of static hMSC culture within MMP-degradable and non-degradable PEG hydrogels. a** Microscopy images of osteogenic markers on day 30, including cell morphology determined by confocal microscopy (MIP, scale bars: 100 μm), collagen fiber secretion determined by Picrosirius-polarization microscopy (scale bars: 100 μm), matrix mineralization determined by Alizarin red staining (scale bars: 100 μm) and osteocalcin expression by immunofluorescence staining (MIP, scale bars: 50 μm). **b** Quantification of collagen content by fiber hue depending on the matrix degradability at day 8 and day 30, color indicates fiber thickness from green (thin, immature) to red (thick, mature), $n = 3$ samples. **c** Quantification of red (mature) fiber content, $n = 3$ samples (mean ± SD, two-way ANOVA/Tukey). **d** Quantification of Alizarin red staining (ARS) area fraction indicating the extent of matrix mineralization, $n = 3$ samples (mean ± SD, two-way ANOVA/Tukey). **e** Quantification of osteocalcin expression by mean fluorescence intensity normalized to cell number, $n = 3$ samples (mean ± SD, two-way ANOVA/Tukey).

expression of osteocalcin was observed in the non-degradable hydrogels (Fig. 6e). Together, these results suggest that the MMP-degradable microporous PEG hydrogels are permissive for the formation of 3D bone cell networks and allow for subsequent osteogenic differentiation under static culture conditions.

**On-chip integration of microporous PEG hydrogels**
For potential microfluidic applications, we injected the void-forming hydrogels into a microfluidic chip and introduced perfusion through the porous matrix. Flow visualization showed that interstitial liquids could pass through the microporous PEG hydrogels when applying a pressure gradient on a microfluidic setup (Fig. 7a, b). The pressure difference across an acellular PEG hydrogel was created by applying dissimilar flow rates ($Q_1 > Q_2$) to the two medium channels. The perfusion of different fluorescein isothiocyanate (FITC)-dextran tracer molecules (70 and 500 kDa) was visualized by fluorescence microscopy. Time-lapsed microscopy images (Supplementary Movie 4) indicate that fluorescence intensity increased gradually for both tracers over time. The intensity of tracer molecules increased until reaching a plateau (Supplementary Fig. 15). Yet, no statistical

difference in the intensity change was observed between the two tracer molecules. We thus reason that pores within PEG hydrogels were large enough for both sizes of tracer molecules to pass through. These results correlate with the pore size quantification (Fig. 3) that pore sizes in PEG hydrogels are in the order of micrometers, which are much larger than the sizes of tracers.

To quantify hydrogel permeability, we employed a microfluidic setup (Fig. 7c) to create a pressure difference across the hydrogel caused by different volumes of medium added to each syringe barrel. The change in pressure difference over the hydrogel was monitored and an exponential decay function was fitted (Fig. 7d). The values for exponent coefficient $c$ (in the equation in Supplementary Method− *Quantification of permeability*) and calculated Darcy's permeability $K$ are summarized in Fig. 7e. The addition of 500 kDa dextran increased the permeability of the microporous hydrogels compared to 40 kDa dextran given the larger pore sizes as shown in Fig. 3. Importantly, the permeability is comparable to that of a reference hydrogel made of collagen type I. We envisage that these microporous PEG hydrogels hold great potential for on-chip perfusion cultures using a chemically defined matrix.

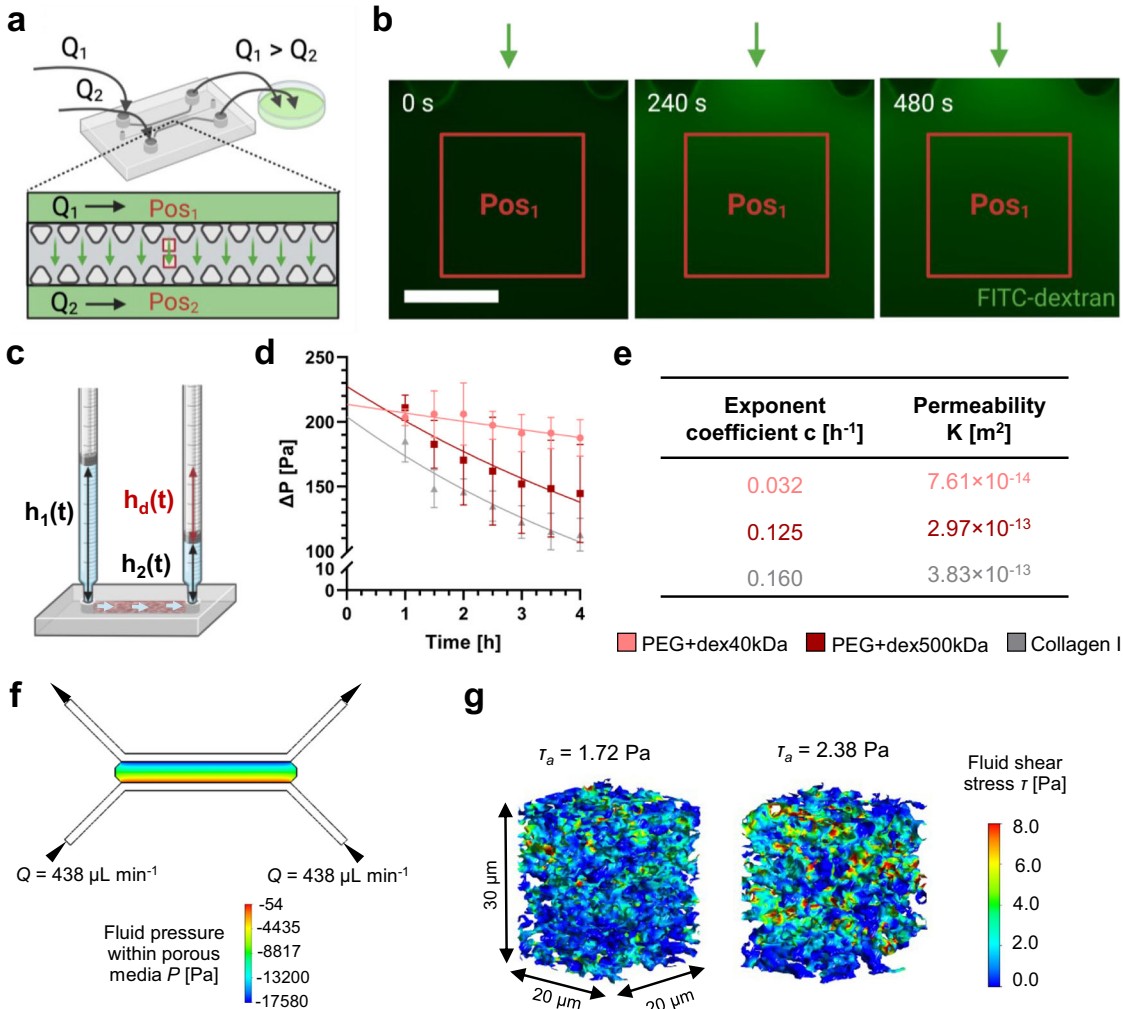

**Fig. 7 | Integration of PEG hydrogels with a microfluidic chip for perfusion of the microporous matrix. a** Microfluidic setup to visualize fluid flow through PEG hydrogels, $Q_1$ and $Q_2$ denote volumetric flow rate, $P_1$ and $P_2$ represent positions of image acquisition. **b** Time-lapsed fluorescence microscopy images of FITC-dextran (500 kDa) tracer perfusing through PEG hydrogels on chip in response to a pressure gradient, scale bar: 200 µm. **c** Microfluidic setup for permeability test depending on a gravity-driven fluid flow through the PEG hydrogels, $h_1$ and $h_2$ denote measured heights, $h_d$ is the height difference to calculate the changes in pressure drop over time. **d** Calculated pressure change over time (symbols) and fitted exponential functions (lines) across 2 PEG hydrogels (40 kDa and 500 kDa dextran) and a reference hydrogel made of 2 mg mL$^{-1}$ collagen type I, $n = 3$ samples (mean ± SD). **e** Resulting Darcy's permeability of the 3 hydrogels obtained from pressure change over time. **f** Pressure distribution within the porous media (homogenized scaffold domain) under the applied flow rate of 438 µL min$^{-1}$ per inlet. **g** Fluid shear stress (FSS) distribution and average FSS ($\tau_a$) within two representative subsections ($x$–$y$–$z$: 20 × 20 × 30 µm) under an applied flow rate of 438 µL min$^{-1}$ per inlet. Illustrations in panels **a** and **c**, created with BioRender.com, were released under a Creative Commons Attribution-NonCommercial-NoDerivs 4.0 International license.

We applied a multiscale and multiphase computational fluid dynamics (CFD) model[44] to estimate the fluid shear stress (FSS) within the microporous hydrogels (Supplementary Fig. 16). Using the permeability values of PEG hydrogels, a global model on the microfluidic chip was generated to estimate the pressure gradient across the hydrogel region in response to a defined flow rate. This pressure gradient was used for defining the loading conditions of the model on a hydrogel subsection. To determine the flow rate for generating a physiologically relevant average FSS of $\tau_a = 2\,\text{Pa}$[7] within the porous space, we conducted a loading (i.e., applied flow rate) variation study using the CFD model based on the pore imaging and permeability data of PEG hydrogels with 500 kDa dextran. Consequently, a flow rate of $Q = 438\,\text{µL min}^{-1}$ was identified as optimal, producing a pressure drop of 256 Pa across a hydrogel subsection located in the channel center (Fig. 7f), resulting in an average FSS of approximately 2 Pa across the hydrogel. Figure 7g depicts FSS distribution in two representative subsections. We used this flow rate to perform an osteogenic on-chip culture for up to 21 days. The estimation of FSS within microporous PEG hydrogels during perfusion serves as a valuable approach to approximate the mechanical cues experienced by embedded cells under dynamic culture conditions. It is important to note that this CFD model does not account for the volumes of embedded cell networks nor cell-secreted nascent proteins on hydrogel permeability. Additionally, hydrogel deformation under perfusion is not considered. For a deeper understanding, future studies should unravel the dynamics of cell-matrix interplay and flow-induced hydrogel deformation using a fluid-structure interaction model.

### On-chip culture of 3D bone cell networks

We investigated the feasibility of generating 3D bone cell networks on a chip. Figure 8a shows hMSC embedded in both degradable and non-degradable matrices during the second day of osteogenic culture. Remarkably, similar to off-chip cultures, hMSC exhibited a remarkable capacity for rapid spreading, forming intricate 3D cell networks with elongated dendritic protrusions within the degradable matrices (Supplementary Movie 5).

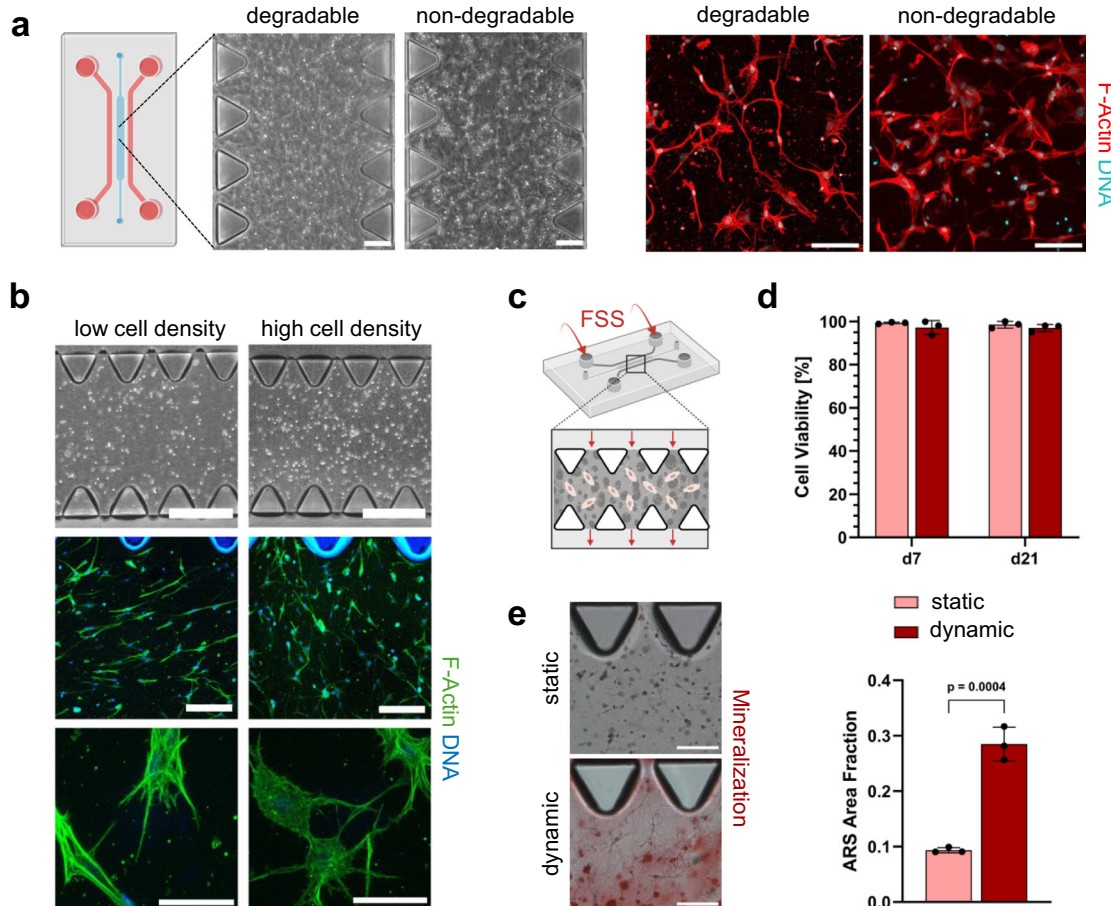

**Fig. 8 | 3D microfluidic culture of hMSC within PEG hydrogels. a** Representative phase contrast (scale bars: 200 μm) and confocal microscopy images (scale bars: 100 μm) showing cell network formation of hMSC embedded in degradable and non-degradable PEG hydrogels on chip after 2 days of osteogenic culture ($n = 3$ samples). **b** Representative phase contrast and confocal microscopy images showing the effect of high ($1 \times 10^6$ mL$^{-1}$) and low ($5 \times 10^5$ mL$^{-1}$) cell seeding density on hMSC morphology inside MMP-degradable PEG hydrogels on day 7 of osteogenic culture on chip, scale bars: top−500 μm, middle−200 μm, bottom−50 μm ($n = 3$ samples). **c–e** 21-day hMSC culture within degradable PEG hydrogels on a chip under static and dynamic

(FSS $\tau_a \approx 2$ Pa) conditions. **c** Schematic illustration of applied FSS to microfluidic hMSC culture ($Q = 438$ μL min$^{-1}$ per inlet). **d** Cell viability on days 7 and 21 of hMSC embedded in MMP-degradable PEG hydrogels on chip in response to FSS or static culture, $n = 3$ samples (mean ± SD, two-way ANOVA/Tukey). **e** Matrix mineralization after 21 days in osteogenic culture determined by Alizarin red staining. Left: microscopy images, scale bars: 200 μm; right: quantification of stained area fraction, $n = 3$ samples (mean ± SD, two-sided Student's t-test). Illustrations in panels **a** and **c**, created with BioRender.com, released under a Creative Commons Attribution-NonCommercial-NoDerivs 4.0 International license.

Cell seeding density has been shown as a key factor in dictating 3D differentiation of primary hOB within a collagen type I hydrogel[29]. We further studied how different cell seeding densities (low: $5 \times 10^5$ mL$^{-1}$ and high: $1 \times 10^6$ mL$^{-1}$) influence the formation of 3D bone cell networks in degradable microporous PEG hydrogels on a chip (Fig. 8b). Following 7 days of cultivation in osteogenic medium, hMSC cultivated at high seeding density exhibited a dendritic morphology, actively forming an intricate 3D cell network. Conversely, cells at low seeding density appeared elongated, yet failed to establish intercellular connections.

Given their microporosity, permeability, and permissiveness for differentiation, we lastly examined the potential of PEG hydrogels for dynamic cell culture on a chip. Initially, we studied the effect of controlled delivery of FSS on the viability of hMSC embedded in microporous hydrogels with 40 kDa dextran. Cells were subjected to perfusion using either a low flow rate ($Q_{low} = 10$ μl min$^{-1}$ per inlet) or high flow rate ($Q_{high} = 100$ μl min$^{-1}$ per inlet), resulting in an average FSS of $\tau_a = 1.74$ Pa and $\tau_a = 17.43$ Pa, respectively, according to CFD simulations (Supplementary Fig. 17). Live/dead assay results on day 13 revealed comparable cell viability between static and low FSS culture, while high FSS led to a significant reduction in cell viability (Supplementary Fig. 18), likely attributed to the high shear stresses well

beyond the physiological range within the LCN porosities in vivo (0.8−3.0 Pa)[7].

Building on these findings, we used microporous PEG hydrogels for a 21-day bone-on-chip perfusion culture. For this, we utilized PEG hydrogels with higher permeability (500 kDa dextran) and the estimated flow rate from the CFD model to apply a physiological average FSS of $\tau_a \approx 2$ Pa (Fig. 7g) for embedded hMSC. Perfusion was applied 3 times per week for 10 min each (dynamic, Fig. 8c) and lasted for 3 weeks with static on-chip cultures as controls. Live-dead staining (Fig. 8d, Supplementary Fig. 19) showed high cell viability in both conditions (static and dynamic) on days 7 and 21 of culture. No significant differences were observed between the groups over time. Immunofluorescence staining for the osteoblast marker osteocalcin[3] revealed similar expression levels on day 21 for both conditions, but a significant increase from day 7 to day 21 for dynamic cultures (Supplementary Fig. 20a, b). Podoplanin, as a marker for dendrite formation of embedding osteocytes[41], was observed on day 7 in dynamic cultures but not in static ones. Although expression levels were higher in static conditions on day 21, podoplanin expression in dynamic cultures appeared more confined within the dendrites. FSS has not only been shown to enhance in vitro osteogenic differentiation but also to induce matrix mineralization[7,45]. Here, we assessed mineral formation in both

static and dynamic hMSC cultures within PEG hydrogels after 21 days (Fig. 8e). Alizarin red staining showed significantly enhanced mineral deposition in the dynamic over static cultures. Together, this proof-of-concept experiment highlights the potential of microporous PEG hydrogels for dynamic cell culture on a chip.

In conclusion, we have successfully developed a synthetic biodegradable microporous hydrogel that effectively supports the 3D culture of human bone cell networks from primary cells. By adjusting the porous architecture and biodegradability of the hydrogels, we investigated 3D cell-matrix interactions such as cell spreading, as well as the resulting influence on in vitro osteogenic differentiation. Within the MMP-degradable matrices, single bone cell precursors respond to the porous architecture, forming an interconnected cell network in 3D and differentiating into an osteoid-like tissue. An essential advantage of this technique lies in the synthetic nature of these hydrogels, which are chemically defined and have minimal batch-to-batch variations. Furthermore, the use of this synthetic non-fibrillar hydrogel facilitates down-stream analysis of cell-secreted matrix proteins such as collagen type I fibers as a potential biomarker for studying (patho-)physiological processes linked to bone formation. Moreover, the unique process of PIPS allows for the formation of interconnected pores in the presence of living cells, a feature unachievable with other types of microporous hydrogels formed through previously described methods[18,20,23]. The resulting permeability allows for fluid flow through the matrix, making it compatible with microfluidic platforms and has the potential to deliver mechanical stimuli to enhance osteogenesis. This in vitro platform may open up new avenues to study bone development with minimal reliance on animal experimentation.

## Methods

### Hydrogel preparation

To prepare PEG hydrogels, 4-PEG-VS (20% w/v in HEPES), 40 kDa or 500 kDa dextran (Sigma-Aldrich, 31389-25G or 31392-10G, in phosphate-buffered saline (PBS) pH 7.4), RGD (China Peptides, N-C: CG**RGD**SP, in PBS pH 6), HA (Sigma-Aldrich, 9067-32-7, 0.5–1% w/v in Hanks' Balanced Salt Solution (Gibco, 14025-050) or Dulbecco's modified Eagle's medium (DMEM, Gibco, high glucose)) and non-degradable crosslinker (PEG-2-SH (2 or 3.4 kDa, Laysan Bio, in PBS pH 6) or MMP-degradable (China Peptides, N-C: KCGPQGIWGQCK or GCRDGPQGIWGQDRCG, in PBS pH 6 or 0.3 M TEOA pH 8))) were thoroughly mixed in this order to ensure PIPS, efficient crosslinking and even distribution of RGD. RGD and crosslinker stock solutions were prepared directly before mixing the precursor solution and kept on ice to prevent rapid oxidation of thiol groups. The final concentrations of each component are noted in the figure captions. Typical ranges were 2.0–2.5% w/v 4-PEG-VS, 0.0–2.0% w/v dextran, 0.25–0.83% w/v HA and a thiol/ene ratio of 0.8 or 1.25 between 4-PEG-VS and crosslinker and 0.07 between 4-PEG-VS and RGD. The exact composition for each experiment is shown in Supplementary Table 1. Once the hydrogel precursor solution was prepared, it was rapidly cast into either a custom-made round poly(dimethylsiloxane) (PDMS) mold on a confocal dish (VWR, 734-2905) or into the central hydrogel channel of a 3D cell culture chip (AIM Biotech, DAX-1) inside the corresponding chip holder (AIM Biotech, HOL-1 or HOL-2) according to the manufacturer's protocol. Hydrogels were crosslinked at 37 °C and 5% $CO_2$ for at least 60 min. After crosslinking, hydrogels were washed 1–2 times with PBS. For the labeled hydrogels, rhodamine-labelled 4-PEG-VS was used instead.

### Characterization of porosity

For a quantitative analysis of pore size and pore connectivity, hydrogels with rhodamine-labeled 4-PEG-VS were prepared, washed with PBS, and incubated at 37 °C to allow for swelling for at least 24 h. Hydrogels were then imaged by confocal microscopy using a Leica SP8 confocal laser scanning microscope with ×25 water immersion objective. Z-stacks of 20–30 µm were obtained and deconvoluted using

Huygens Professional Software. These images were then processed with the MATLAB algorithm developed by Vandaele et al.[38] that uses image segmentation and approximation of pores by spheres to quantify pore size, connectivity and other determinants of the porous architecture. An adaptive threshold sensitivity of 0.6 was chosen. Pore size (external pore radius in 3D), pore connectivity (number of neighboring pores), and porosity were obtained. Pores with external radii above 1 µm were considered for quantification. PEG hydrogels with 0% dextran exhibiting negligible phase separation and featuring sub-micron pores, were omitted from quantification due to the confounding effect of noise affecting the deconvolution of their confocal images.

### Quantification of cell morphology

To quantitatively compare the mean cell area between groups, confocal microscopy (Leica SP8) z-stacks of 100–200 µm were acquired with a ×25 water immersion objective. Using the MIP of these images, the mean cell area was calculated by determining the area of actin in Fiji/ImageJ and dividing it by the number of nuclei in the same image. To analyze the dendrite length of cells embedded within microporous hydrogels, MIPs (100 µm) of confocal microscopy images of actin-stained samples were used. Images were processed using the ImageJ plugin NeuriteQuant[40] to segment cells into cell bodies and dendrites and subsequently quantify the mean length of all dendrites of a single cell.

### Collagen imaging and quantification

Cell-secreted collagen was investigated using Picrosirius red staining (Sigma-Aldrich, 365548). Briefly, the cryosections were stained in picrosirius red (0.1% in saturated aqueous picric acid) for 1 h and washed in two changes of acidified water. Polarized light microscopy images (polarization filter angles at 0° and 90°) were taken in transmission mode with a Zeiss AxioImager.Z2 at ×20 magnification. Collagen content was then quantified according to the method described by Rich and Whittaker[43]. First, images were converted to 8-bit RGB. The color threshold in the Hue spectrum was next split into the different colors as: red 2–9 and 230–256, orange 10–38, yellow 39–51 and green 52–128. The pixel area of each color was measured. The different hue ranges measured were expressed as a percentage of all pixels in the image. Fiber thickness and maturity of collagen fiber increases from green, yellow, orange to red.

### Visualization of fluid flow on-chip

To visualize fluid flow through the porous matrix, the hydrogel was cast into a microfluidic chip (AIM Biotech, DAX-1) with attached luer connectors (AIM Biotech, LUC-1). After hydration for 24 h, flow imaging was performed on a wide-field microscope (Olympus, IX83). Two different tracer solutions were prepared by diluting a 0.1% w/v stock solution of 70 kDa or 500 kDa FITC-dextran (Sigma-Aldrich, FD70S-100MG and FD500S-100MG) 1:1000 in DMEM. The chips were then mounted onto the microscope stage, the tracer solutions were filled into 20 mL syringes which were connected to the luer connectors on the chips via needles (0.80 × 22 mm, blunt, Braun), tubing (Semadeni Plastics, 4348) and male luer connectors (Cole Parmer, 45518-07) that were primed with liquid to prevent air from being trapped along the flow path. Using 2 syringe pumps (WPI, AL-1000), an interstitial flow was created by applying different flow rates ($Q_1 = 334$ µL min$^{-1}$ and $Q_2 = 10$ µL min$^{-1}$) to the medium channels. The perfusion of the PEG hydrogels with tracer molecules was imaged in two different positions using a filter for FITC and a ×20 air objective every 20 s for 8 min. The pumps were switched on between time-points 2 and 3. For quantification, the mean intensity in a rectangular area at positions $Pos_1$ and $Pos_2$ was measured in Fiji/ImageJ for each time-point and both tracers. This intensity was normalized by the mean intensity at time-point 0 for each tracer and position.

## Statistics

Statistical analysis was performed in GraphPad Prism 8.2.0. Depending on the number of variables, ordinary one-way or two-way analysis of variance (ANOVA) was used followed by Tukey's test for multiple comparisons if more than two groups were compared. For the comparison of only two groups, Student's $t$-test (two-sided) was used. Further, $p$ values are indicated in the figures. Data is shown as mean ± SD, and the number of independent samples ($n$) is indicated in the figure caption. Whiskers of box plots indicate minimum and maximum values, upper and lower limits indicate 75th and 25th percentiles, respectively, and the center line indicates the median.

## Reporting summary

Further information on research design is available in the Nature Portfolio Reporting Summary linked to this article.

## Data availability

The data that support the findings of this study are available in the ETH Zurich Research Collection with the identifier https://doi.org/10.3929/ethz-b-000638979.

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

## Acknowledgements

The authors thank Sophie Zengerle, Isabel Hui, Wanwan Qiu, Christian Gehre, Margherita Bernero, Dr. Bregje de Wildt, and Dr. Yinyin Bao for experimental support; Dr. Cecilia Giunta and Prof. Dr. Marianne Rohrbach at the University Children's Hospital Zurich for providing with some of the primary human osteoblasts; and Dr. Tobias Schwarz in the Scientific Center for Optical and Electron Microscopy (ScopeM) of ETH Zurich for technical support. This project was supported by the Swiss National Science Foundation (no. 190345, 206501, 188522, X.H.Q.) and the Swiss State Secretariat for Education, Research and Innovation (no. MB23.00008, X.H.Q.).

## Author contributions

Conceptualization: X.-H.Q., D.Z., M.Z.M. Investigation: D.Z., M.Z.M., M.H., L.B., F.Z., X.-H.Q. Methodology: D.Z., M.Z.M, M.H., L.B., F.Z., P.F., S.S.L., X.-H.Q. Data curation: D.Z., M.Z.M., M.H., L.B., F.Z., X.-H.Q. Formal analysis: D.Z., X.-H.Q. Validation: D.Z., M.H., X.-H.Q. Visualization: D.Z., M.Z.M, L.B., F.Z., X.-H.Q. Writing—original draft: D.Z., X.-H.Q. Writing—review and editing: D.Z., M.Z.M., M.H., L.B., F.Z., P.F., S.S.L., M.Z.-W., R.M., X.-H.Q. Funding acquisition and project administration: X.-H.Q. Supervision: X.-H.Q. and R.M.

## Competing interests

ETH Zurich (with X.-H.Q., D.Z., and R.M.) has filed patent applications for this new in vitro cell culture technology. Correspondence and requests for materials should be addressed to X.-H.Q. Additionally, it is confirmed that no other authors involved have any competing interests.
