## [Peer Review File · Nature Communications]

Reviewers' Comments:

Reviewer #1:

Remarks to the Author:

Manuscript Number: NCOMMS-23-50926

Title : Synthetic Biodegradable Void-forming Hydrogels for In Vitro 3D Culture of Functional Human Bone Cell Networks

In this manuscript, the authors report a synthetic biodegradable macroporous hydrogel for efficient formation of 3D networks from human primary 24 cells, analysis of cell-secreted extracellular matrix (ECM) and microfluidic integration. In general, the topic is interesting. Basically, this manuscript was written in a reasonable organizational framework, and would be helpful to the readers in the field. The manuscript emphasizes void-Forming PEG Hydrogels and provides some useful information in the topic area, but the fundamental principles for in situ pore formation by polymerization-induced phase separation should be elucidated by more sentences. Although Fig.1 and Fig.2a have the schematic representation, more deep elucidation may be needed to impress readers. And there are also other issues that need to be addressed as follows.

In section of introduction, there is a sentence "For 3D embedding, a variety of natural a variety of natural (e.g., collagen, [7] gelatin [8]) and synthetic hydrogels (e.g., clickable polyvinyl alcohol [9] or poly(ethylene glycol) (PEG) [10]) have been reported. However, these hydrogels suffer from small pores (5-50 nm) and low permeability". Actually, some type of semi-synthetic hydrogels including the hyaluronic acid or dextran-based hydrogels have the tunable sizes of the inner pores that can be larger than 50 nm, which may have moderate permeability. It seems like that some research papers reporting the synthetic hydrogels whose inner microstructures can be rationally designed have been missed. The inclusion of additional relevant literature citations is recommended to enhance the comprehensiveness of this topic review.

In section 2.1, could the authors provide the complete sequence of the arginyglycylaspartic acid peptide including the grafted functional groups if there were? The duration time for crosslinking process appears to be relatively longer, and could the authors explain why the crosslinking time takes up to one hour since crosslinking time of some hydrogels including dextran or HA needs less than 30 mins ?

In section 2.2, there is a sentence "Varying the dextran molecular weight (MW, 40 kDa or 500 kDa), however, did not significantly affect the hydrogel mechanical properties (Supplementary Figure 2b)." Could the authors explain why the dextran molecular weight does not significantly affect the hydrogel mechanical properties, since the hydrogel modulus usually changed when the dextran molecular weight significantly changed?

In the caption of Fig. 2a, the sentence "upon addition of a matrix metalloproteinase (MMP)-degradable di-cysteine crosslinker, 4-arm PEG vinyl sulfone (4-PEG-VS) is crosslinked in the presence of dextran and hyaluronic acid (HA)" mentions the hyaluronic acid (HA), yet the authors did not explain the function of HA in this schematic diagram.

In section 2.3, the authors wrote "the increase of dextran concentration from 0% to 0.5%, 1% and 2% led to an increase in pore radius". Yet this result needs a reasonable explanation, since the higher concentration of polymer molecule usually leads to a larger density locally, which possibly result in a smaller pore radius.

Figure 4c presents the quantification of mean cell area for hMSCs, while figure 4h presents the quantification of mean cell area for human osteoblasts (hOBs). But the caption for figure 4h does not contain the information on cell type. The authors use the one-way ANOVA/Tukey for statistics

in figure 3b-c, but use the two- way ANOVA/Tukey in figure 4c. Could the authors explain the reason for using different methods for statistics?

In section 2.7, the authors further studied how different cell seeding density (low: $5 \times 10^5 \text{ ml}^{-1}$ and high: $1 \times 10^6 \text{ ml}^{-1}$) influences the formation of 3D bone cell networks on chip. However, the cell density range was specifically selected here, and the difference between the cell densities may be not adequate. If it's convenient, It is suggested the cell density larger than $1 \times 10^6 \text{ ml}^{-1}$ (such as $2 \times 10^6 \text{ ml}^{-1}$ or $3 \times 10^6 \text{ ml}^{-1}$) be also applied for the experiment and check if the 3D bone cell networks would normally formed or other types of morphologies would emerge.

For confocal microscopy, the model type of the instrument is suggested to be provided. Why was the 25× objective chosen for imaging? not other magnification for the objective? Does it influence the image quality for specific samples?

In the section 4, the protocol set a thiol/ene ratio of 0.8 or 1.25 between 4-PEG-VS and crosslinker and 0.07 between 4-PEG-VS and RGD. Could authors elaborate what the advantages of utilizing these ratio values are?

The section number 4 should be revised as 3, because there is no section 3 above the section 4.

There are a few typos or grammar mistakes in the manuscript. The typos and grammar mistakes need to be checked and corrected through the whole manuscript.

Reviewer #2:

Remarks to the Author:

Zauchner, et al. describe a method for preparing hydrogel constructs with cell-scale pores through a single polymerization step that leverages polymer phase separation. This is an elegant approach for forming hydrogels with void spaces through which cells can extend processes and migrate, which may be useful for a range of applications in tissue engineering, regenerative medicine, and organ-on-a-chip systems. The authors further demonstrate that introduction of MMP-cleavable crosslinks into the gels enabled robust cell spreading, deposition of new ECM, and upregulation of osteogenic markers. The authors also demonstrate that their gels can be loaded into microfluidic chips and support cell growth under perfused conditions.

1. Regarding the characterization of pore formation in the gels, the authors indicate in figure 3 that the 0% dextran conditions tested still exhibit pores with radii on the order of μm in size. The fluorescence micrographs do not seem to show significant phase separation and void formation in the absence of dextran. Do the authors believe they are getting phase separation even in the absence of dextran (for instance, due to the presence of HA)? If there is not substantial phase separation, the pore size in the 0% dextran gels would be expected to be on the order of nm. Have the authors tested this condition in their fluorescent tracer perfusion experiment (figure 6) to demonstrate that this condition also has large scale pores?

2. The authors report the bulk storage and loss moduli for the gels, but cells interfacing with these materials would not sense the macroscopically determined moduli, but rather the local moduli for the crosslinked gel regions to which they are adherent. Given that osteogenesis by MSCs and osteoblasts is well-known to be regulated by substrate mechanical cues, have the authors characterized the mechanics of the crosslinked phase only (e.g., by micro or nanoindentation)? This may be different than a non-phase separated bulk gel with the same starting PEG concentration, as the authors acknowledge that the phase separation process may result in increased PEG content in the phase separated regions. If mechanical properties vary substantially among conditions, this could prove to be a confounding factor when reading out metrics of osteogenic fate.

3. For the rheological characterization of the gels, the authors point out that there is a substantial viscous character to the networks. They speculate that the HA in the system may contribute to the ability of the materials to relax applied stresses. Have the authors tested the stress relaxation rates in these materials? And would this viscoelastic character be expected to remain consistent throughout the culture duration, or can the HA diffuse out of the gels and thereby result in more elastic character over time? As a minor point, it is not clear from the Figure 2e caption what material composition was tested in the figure panel shown.

4. To demonstrate the osteogenic potential of the embedded cells, the authors report the fraction of area of histological sections that stained positive for mineral deposition and osteocalcin. In these images, it looks like the cell number is also quite different between the degradable and non-degradable materials. Are there differences in cell proliferation between the degradable and non-degradable materials? To account for differences in cell number in the imaged sections, the authors should also present the mineralization and osteocalcin-positive areas normalized by a metric of cell number (e.g., per number of nuclei or per DAPI-positive area). Normalization to cell number will show that the increase in osteogenic signature in the degradable materials is not only due to proliferation differences, but rather to altered ability to spread and form cellular networks.

5. A major advantage of the gel system cited by the authors is the ability to incorporate the gels into microfluidic organ-on-a-chip systems. The authors show proof of principle that cells encapsulated in these gels can survive physiological levels of shear stress in the microfluidic devices. However, if the authors' goal of a "bone-on-a-chip" is to be realized, the cells need to be able to undergo osteogenic differentiation. Thus, markers of osteogenic differentiation in the dynamic (shear stress) culture conditions should also be assessed.

6. The authors' assertion that the PIPS method uniquely allows for pore formation in the presence of live cells is not accurate based on the papers cited to support this statement (lines 279-281). For instance, ref. 16 has pores form while live cells are present due to hydrolytic degradation of a sacrificial porogen phase. The system in the present manuscript does have the advantage of requiring only a single processing step (phase separation during crosslinking) in order to generate the porous structures.

Reviewer #3:

Remarks to the Author:

The manuscript reports the engineering of MMP-sensitive PEG hydrogels for supporting bone formation. The work presents an interesting study design but lacks the novelty and impact in respect to performance of the proposed hydrogels. Moreover, it is not clear how the synthetic hydrogel performs as compared to the gold standard material. Several important techniques are missing in order to better characterise the hydrogels in respect to bioactivity and osteogenic ability. The Introduction section should be improved to better reflect the current state of the art. Figure 1 schematics are not scientifically accurate and is too oversimplistic. The degradation of the synthetic hydrogels should be carried out at different pH's. The cell viability and proliferation should be performed until longer culture times (e.g. 14 days). Different osteogenic markers should be quantitatively investigated at early and late stage by RT-PCR, and gold standard materials should be used as controls (e.g. hydroxyapatite). Mineralization and osteogenic markers should be quantitatively investigated in the 3D microfluidic culture of cell-laden PEG hydrogels.

REVIEWER COMMENTS

Reviewer #1 (Remarks to the Author):

In this manuscript, the authors report a synthetic biodegradable macroporous hydrogel for efficient formation of 3D networks from human primary cells, analysis of cell-secreted extracellular matrix (ECM) and microfluidic integration. In general, the topic is interesting. Basically, this manuscript was written in a reasonable organizational framework, and would be helpful to the readers in the field.

Authors' response: We are grateful for the reviewer's positive feedback and constructive suggestions, which are crucial to improve this manuscript.

1. The manuscript emphasizes void-Forming PEG Hydrogels and provides some useful information in the topic area, but the fundamental principles for in situ pore formation by polymerization-induced phase separation should be elucidated by more sentences. Although Fig.1 and Fig.2a have the schematic representation, more deep elucidation may be needed to impress readers. And there are also other issues that need to be addressed as follows.

Authors' response: We thank the reviewer for this constructive comment. We have added a detailed description of polymerization-induced phase separation (PIPS) in the main text (**Section 2.1.**):

“Polymerization-induced phase separation (PIPS) is a dynamic process where an initially miscible mixture (single-phase) undergoes phase decomposition during polymerization of the reactive components, thereby forming a multi-phase blend. Upon mixing the components (4-PEG-VS, HA, dextran and crosslinker) and elevating the temperature to 37 °C, the mixture undergoes PIPS induced by Michael addition crosslinking (Figure 2a) and forms PEG hydrogels with interconnected porosity through a single processing step.³⁴”

2. In section of introduction, there is a sentence “For 3D embedding, a variety of natural (e.g., collagen, [7] gelatin [8]) and synthetic hydrogels (e.g., clickable polyvinyl alcohol [9] or poly(ethylene glycol) (PEG) [10]) have been reported. However, these hydrogels suffer from small pores (5-50 nm) and low permeability”. Actually, some type of semi-synthetic hydrogels including the hyaluronic acid or dextran-based hydrogels have the tunable sizes of the inner pores that can be larger than 50 nm, which may have moderate permeability. It seems like that some research papers reporting the synthetic hydrogels whose inner microstructures can be rationally designed have been missed. The inclusion of additional relevant literature citations is recommended to enhance the comprehensiveness of this topic review.

Authors' response: We appreciate your constructive suggestion. Therefore, we have revised the Introduction with a more comprehensive overview of the current state-of-the-art in this field (**Section 1, paragraph 2**):

“For 3D embedding, a variety of natural (e.g., collagen⁸, gelatin⁹) and synthetic hydrogels (e.g., clickable polyvinyl alcohol¹⁰ or poly(ethylene glycol) (PEG)¹¹) have been reported. However, these hydrogels often have nanoscale pore sizes (5–100 nm) and limited permeability.¹² Consequently, cell spreading relies on matrix degradation via hydrolysis or proteolysis through cell-secreted matrix metalloproteinases (MMPs). In contrast, top seeding typically relies on scaffolds with large pores (100–600 μm)^{13–16} where the cell to surface interface is 2D. As a result, cells often fail to form 3D networks. Various techniques, such as emulsification¹⁷, porogen leaching^{18–21} and particle or microgel annealing^{22–26} have been employed to create microporous hydrogels with relevant pore sizes for cell spreading of 5–150 μm . Nevertheless, most of these methods have limitations in generating interconnected pores in the presence of living cells, while others require the degradation of a sacrificial porogen phase and therefore multiple processing steps.”

3a. In section 2.1, could the authors provide the complete sequence of the arginylglycylaspartic acid peptide including the grafted functional groups if there were?

Authors’ response: As detailed in Section 2.1., the complete sequence of the arginylglycylaspartic acid peptide is CGRGDSP (N-C: Cys-Gly-Arg-Gly-Asp-Ser-Pro). This peptide provides a thiol group reacting with the vinyl sulfone groups of 4-arm-PEG-VS through thiol-Michael addition, as illustrated in Figure 2a.

Figure 2. Synthesis and characterization of microporous PEG hydrogels by polymerization-induced phase separation (PIPS). **a)** Illustration of in situ pore formation by PIPS at 37°C: upon addition of a matrix metalloproteinase (MMP)-degradable di-cysteine crosslinker, 4-arm PEG vinyl sulfone (4-PEG-VS) is crosslinked in the presence of dextran and hyaluronic acid (HA), leading to in situ pore formation.

3b. The duration time for crosslinking process appears to be relatively longer, and could the authors explain why the crosslinking time takes up to one hour since crosslinking time of some hydrogels including dextran or HA needs less than 30 mins?

Authors' response: Thank you for this comment. The decision to set the crosslinking time for the hydrogel at 60 minutes was based on observations that, for most compositions, the storage modulus G' plateaued at 60 minutes. It is important to note, however, that the actual crosslinking process - marked by the crossover of the loss modulus (G'') and the storage modulus (G') - occurs at a significantly quicker pace, as shown in **Response Figure 1**.

Response Figure 1. Storage (G') and loss (G'') modulus during crosslinking of PEG hydrogel (2.5% 4-PEG-VS, 1% 40 kDa dextran, non-degradable crosslinker).

Accordingly, the main text in **Section 2.1.** and **Section 2.2.** was adapted to clarify this point:

“During crosslinking, spatial patterns of binodal nucleation were observed (Figure 2b, Supplementary Movie 1).”

“Depending on the hydrogel composition, gelation (crossover of G' and loss modulus (G'')) started immediately upon in situ crosslinking at 37°C and plateaued after 60 min.”

4. In section 2.2, there is a sentence “Varying the dextran molecular weight (MW, 40 kDa or 500 kDa), however, did not significantly affect the hydrogel mechanical properties (Supplementary Figure 2b).” Could the authors explain why the dextran molecular weight does not significantly affect the hydrogel mechanical properties, since the hydrogel modulus usually changed when the dextran molecular weight significantly changed?

Authors' response: While we anticipated a change in the G' -plateau after 60 minutes of crosslinking as a result of the variation of the molecular weight of dextran, such differences were not evident in our rheological measurements ($p > 0.05$, **Supplementary Figure 2b**). We assume that the molecular

weights of dextran (i.e., 40 kDa and 500 kDa) do not significantly influence the local concentration of crosslinkable components, which strongly influence the bulk hydrogel modulus.

Supplementary Figure 2b. Storage modulus (G') of PEG hydrogels with low (40 kDa) and high (500 kDa) dextran Mw after 60 min of crosslinking at 37 °C, $n=3$ (Student's t-test).

5. In the caption of Fig. 2a, the sentence “upon addition of a matrix metalloproteinase (MMP)-degradable di-cysteine crosslinker, 4-arm PEG vinyl sulfone (4-PEG-VS) is crosslinked in the presence of dextran and hyaluronic acid (HA)” mentions the hyaluronic acid (HA), yet the authors did not explain the function of HA in this schematic diagram.

Authors' response: Thank you for bringing this to our attention. We have adapted the diagram in **Revised Figure 2a** accordingly:

Revised Figure 2. Synthesis and characterization of microporous PEG hydrogels by polymerization-induced phase separation (PIPS). **a)** Illustration of in situ pore formation by PIPS at 37°C: upon addition of a matrix metalloproteinase (MMP)-degradable di-cysteine crosslinker, 4-arm PEG vinyl sulfone (4-PEG-VS) is crosslinked in the presence of dextran and hyaluronic acid (HA), leading to in situ pore formation.

Additionally, we added an explanation to the **Abstract** and the main text in **Section 2.1.:**

“Pore formation is triggered by thiol-Michael addition crosslinking of a viscous precursor solution supplemented with hyaluronic acid and dextran.”

“HA is necessary to increase the complex viscosity in a suitable range (0.3–0.6 Pa s) and thus prevent the phases from collapsing during PIPS.”

6. In section 2.3, the authors wrote “the increase of dextran concentration from 0% to 0.5%, 1% and 2% led to an increase in pore radius”. Yet this result needs a reasonable explanation, since the higher concentration of polymer molecule usually leads to a larger density locally, which possibly result in a smaller pore radius.

Authors’ response: Thank you for pointing this out. We agree with the reviewer that the higher concentration of polymer molecule may lead to a larger density locally and a smaller pore radius. However, in the present study, we only screened the low range of dextran concentrations (0-2%) so that the mixture is initially miscible before PIPS. We expected that only the PEG-rich phase undergoes crosslinking, leaving the dextran phase non-crosslinked. In the selected range (0-2% dextran), a higher concentration of dextran leads to larger pore sizes based on the pore imaging data using rhodamine-labelled 4-PEG-VS (**Figure 3a**).

Figure 3. Characterization of the porous architecture of void-forming PEG hydrogels in function of dextran concentrations and molecular weights. **a)** Confocal microscopy images of rhodamine-labeled PEG hydrogels formed with varying dextran concentrations without physical constraints, scale bars: 10 μm .

7a. Figure 4c presents the quantification of mean cell area for hMSC, while figure 4h presents the quantification of mean cell area for human osteoblasts (hOBs). But the caption for figure 4h does not contain the information on cell type.

Authors’ response: Thank you for pointing this out. We have revised the caption of **Figure 4f-h** accordingly:

“... **f)** Semiautomatic labelling of hOB cell bodies and dendrites by NeuriteQuant plugin in ImageJ for quantitative comparisons of cell morphologies, scale bars: 100 μm . **g)** Quantification of mean hOB dendrite length per cell

using NeuriteQuant in degradable and non-degradable hydrogels, n=3 (*p<0.05, Student's t-test). **h)** Quantification of mean hOB cell area in degradable and non-degradable hydrogels, n=3 (Student's t-test)."

7b. The authors use the one-way ANOVA/Tukey for statistics in figure 3b-c, but use the two-way ANOVA/Tukey in figure 4c. Could the authors explain the reason for using different methods for statistics?

Authors' response: The choice between one-way and two-way ANOVA for the plots mentioned depends on the structure of the data being analyzed:

- One-way ANOVA/Tukey in **Figure 3b-c**: This method was chosen because the focus here was on comparing different groups. Since there is only one independent variable (the groups), a one-way ANOVA is sufficient to assess whether there are significant differences in the means across these groups.
- Two-way ANOVA/Tukey in **Figure 4c**: The choice of a two-way ANOVA for this figure is due to the presence of two independent variables: time points and groups. This analysis allows us to evaluate not only the effect of each independent variable but also whether there is an interaction effect between the two variables.

8. In section 2.7, the authors further studied how different cell seeding density (low: $5 \times 10^5 \text{ ml}^{-1}$ and high: $1 \times 10^6 \text{ ml}^{-1}$) influences the formation of 3D bone cell networks on chip. However, the cell density range was specifically selected here, and the difference between the cell densities may be not adequate. If it's convenient, it is suggested the cell density larger than $1 \times 10^6 \text{ ml}^{-1}$ (such as $2 \times 10^6 \text{ ml}^{-1}$ or $3 \times 10^6 \text{ ml}^{-1}$) be also applied for the experiment and check if the 3D bone cell networks would normally formed or other types of morphologies would emerge.

Authors' response: Thank you for your comment. We selected and adjusted the cell densities after taking account of a study by Nasello et al.^[1] The authors show cell densities of $1 \times 10^6 \text{ ml}^{-1}$ (high) and $2.5 \times 10^5 \text{ ml}^{-1}$ (low) significantly influence osteogenic differentiation of hOB in a 3D collagen culture. In our study, we have also utilized higher cell densities of $3 \times 10^6 \text{ ml}^{-1}$ with our microfluidic model (as depicted in **Figure 8a** and **8c-e** and **Supplementary Figure 19** and **20**). As anticipated, higher cell densities resulted in even more pronounced cell network formation on chip.

Figure 8. 3D microfluidic culture of hMSC within PEG hydrogels. **a)** Phase contrast (scale bars: 200 μm) and confocal microscopy images (scale bars: 100 μm) showing cell network formation of hMSC embedded in degradable and non-degradable PEG hydrogels on chip after 2 days of osteogenic culture. **b)** Phase contrast and confocal microscopy images showing the effect of high ($1 \times 10^6 \text{ ml}^{-1}$) and low ($5 \times 10^5 \text{ ml}^{-1}$) cell seeding density on hMSC morphology inside MMP-degradable PEG hydrogels on day 7 of osteogenic culture on chip, scale bars: top - 500 μm , middle - 200 μm , bottom - 50 μm . **c-e)** 21-day hMSC culture within degradable PEG hydrogels on chip under static and dynamic (FSS $\tau_0 \approx 2 \text{ Pa}$) conditions. **c)** Schematic illustration of applied FSS to microfluidic hMSC culture ($Q=438 \mu\text{l min}^{-1}$ per inlet). **d)** Cell viability on day 7 and 21 of hMSC embedded in MMP-degradable PEG hydrogels on chip in response to FSS or static culture, $n=3$ (two-way ANOVA/Tukey). **e)** Matrix mineralization after 21 days in osteogenic culture determined by Alizarin red staining. Left: microscopy images, scale bars: 200 μm ; right: quantification of stained area fraction, $n=3$ (** $p < 0.001$, Student's t-test). Illustrations in a) and c) created with BioRender.com.

Supplementary Figure 19. Confocal microscopy images (MIPs) of live-dead stained hMSC after 7 and 21 days of static and dynamic osteogenic culture within degradable PEG hydrogels, scale bars: 200 μm .

Supplementary Figure 20. 21-day hMSC culture within degradable PEG hydrogels on chip with and without the application of FSS $\tau_a \approx 2$ Pa as simulated using the CFD model. **a)** Confocal microscopy images (MIPs) of osteocalcin and podoplanin immunofluorescence staining as osteoblast and early osteocyte markers, respectively, scale bars: 100 μm . **b)** Quantification of fluorescence intensity of immunostaining normalized to cell number for osteocalcin and podoplanin, $n=3$ (* $p<0.05$, ** $p<0.01$, two-way ANOVA/Tukey).

9. For confocal microscopy, the model type of the instrument is suggested to be provided. Why was the 25× objective chosen for imaging? not other magnification for the objective? Does it influence the image quality for specific samples?

Authors' response: Thank you for this comment. We noticed that we specified the instrument type only in the Supplementary Methods section and have adapted the corresponding sections in the main text and Supplementary Methods with more details about the imaging setup:

Section 3., paragraph 2:

“Hydrogels were then imaged by confocal microscopy using a Leica SP8 confocal laser scanning microscope with 25× water immersion objective.”

Section 3., paragraph 3:

“To quantitatively compare mean cell area between groups, confocal microscopy (Leica SP8) z-stacks of 100–200 μm were acquired with a 25× water immersion objective.”

Supplementary Methods, paragraph 7:

“Samples were imaged using confocal microscopy (Leica SP8) with a 10× air objective.”

We have used varying objectives depending on the requirements for specific experiments: usually 10× magnification for overview images, 63× for sub-cellular details and pore visualization and 25× for most other acquisitions.

10. In the section 4, the protocol set a thiol/ene ratio of 0.8 or 1.25 between 4-PEG-VS and crosslinker and 0.07 between 4-PEG-VS and RGD. Could authors elaborate what the advantages of utilizing these ratio values are?

Authors' response: We based our selection of these values on the characteristics of previously studied PEG hydrogels.^[2] Lutolf et al.^[3] demonstrated that a stoichiometric thiol/ene ratio of 1.25 yielded PEG hydrogels with the highest elastic modulus, guiding us to adopt this ratio as our maximum thiol/ene limit to ensure hydrogel stability. We opted for a thiol/ene ratio of 0.8 between the di-thiol crosslinker and 4-PEG-VS to ensure that not all ene groups on the 4-PEG-VS are consumed during in situ crosslinking, leaving sufficient reactive sites to conjugate with thiolated RGD peptides to promote cell adhesion. Both ratios were effective in forming stable adhesive hydrogels. The ratio of RGD peptides was chosen below 0.1 to ensure crosslinking efficiency while providing a sufficient ligand density.

11. The section number 4 should be revised as 3, because there is no section 3 above the section 4.

Authors' response: Thank you. We have corrected this mistake.

12. There are a few typos or grammar mistakes in the manuscript. The typos and grammar mistakes need to be checked and corrected through the whole manuscript.

Authors' response: Thank you for pointing this out. We have carefully checked the manuscript for spelling and grammar mistakes and corrected them accordingly with the support of a native speaker.

Reviewer #2 (Remarks to the Author):

Zauchner, et al. describe a method for preparing hydrogel constructs with cell-scale pores through a single polymerization step that leverages polymer phase separation. This is an elegant approach for forming hydrogels with void spaces through which cells can extend processes and migrate, which may be useful for a range of applications in tissue engineering, regenerative medicine, and organ-on-a-chip systems. The authors further demonstrate that introduction of MMP-cleavable crosslinks into the gels enabled robust cell spreading, deposition of new ECM, and upregulation of osteogenic markers. The authors also demonstrate that their gels can be loaded into microfluidic chips and support cell growth under perfused conditions.

Authors' response: Thank you for your positive feedback and insightful suggestions on our manuscript.

1. Regarding the characterization of pore formation in the gels, the authors indicate in figure 3 that the 0% dextran conditions tested still exhibit pores with radii on the order of μm in size. The fluorescence micrographs do not seem to show significant phase separation and void formation in the absence of dextran. Do the authors believe they are getting phase separation even in the absence of dextran (for instance, due to the presence of HA)? If there is not substantial phase separation, the pore size in the 0% dextran gels would be expected to be on the order of nm. Have the authors tested this condition in their fluorescent tracer perfusion experiment (figure 6) to demonstrate that this condition also has large scale pores?

Authors' response: Thank you for your insightful comment. After reviewing your feedback, we revisited our confocal imaging data (**Figure 3** and **Supplementary Figures 3-4**) and found that the confocal images for the 0% dextran hydrogels indeed did not exhibit notable phase separation. We observed an issue during the image deconvolution process, as illustrated by the comparison of images before (left) and after (right) deconvolution for the 0% dextran sample (**Response Figure 2**). The deconvolution process introduced false pores due to noise, leading to an overestimation of pore sizes in our analysis. Note that the length scales of the signals below hundreds of nanometers are difficult to distinguish due to the diffraction limit. Therefore, we have removed the 0% dextran conditions from our analysis.

Response Figure 2. Confocal microscopy images of rhodamine-labeled PEG hydrogels formed with without dextran before (left) and after (right) deconvolution, scale bars: 10 μm .

As indicated in **Supplementary Figure 1**, we do not anticipate micrometer-scale pores in the 0% dextran hydrogels, which informed our decision to exclude this condition from the fluorescent tracer perfusion experiments. We reasoned that the expected low permeability of these hydrogels would hinder fluid flow, potentially causing pressure buildup and resulting in leakage or displacement of the hydrogel from the chip.

Supplementary Figure 1. Confocal microscopy images of PEG hydrogels formed by Michael addition crosslinking without and with dextran, scale bars: 10 μm .

We have adapted the corresponding sections in the manuscript and removed the 0% dextran conditions:

Section 2.3.:

“...As shown in Figure 3a-c and Supplementary Figure 3, the increase of dextran concentration from 0.5% to 1% and 2% led to an increase in pore radius (median pore radius: 7.0, 10.0 and 10.0 μm , respectively) and porosity (37.6 ± 4.1 , 50.4 ± 1.9 and $55.8 \pm 5.9\%$, respectively) as well as higher pore connectivity. The pore size in hydrogels without dextran was in the sub- μm range and thus too small to be quantifiable. ...”

Revised Figure 3. Characterization of the porous architecture of void-forming PEG hydrogels in function of dextran concentrations and molecular weights. **a)** Confocal microscopy images of rhodamine-labeled PEG hydrogels formed with varying dextran concentrations without physical constraints, scale bars: 10 μm . **b-c)** Quantification of pore radius and porosity of hydrogels formed with varying dextran concentrations, $n=3$ ($***p < 0.001$, $****p < 0.0001$, one-way ANOVA/Tukey). **d)** Confocal microscopy images of rhodamine-labeled PEG hydrogels formed with 1.0% low M_w (40 kDa) and high M_w (500 kDa) dextran without physical constraints, scale bars: 10 μm . **e-f)** Quantification of pore radius and porosity of hydrogels formed with low M_w (40 kDa) and high M_w (500 kDa) dextran, $n=3$ ($*p < 0.05$, $****p < 0.0001$, Student's t-test). Illustrations created with BioRender.com.

Revised Supplementary Figure 3. Characterization of the porous architecture of void-forming PEG hydrogels with varying dextran concentration. **a)** Distribution of pore radii in PEG hydrogels with dextran concentrations ranging from 0.5–2.0%, $n=3$. **b)** Distribution of pore connectivity in PEG hydrogels with dextran concentrations ranging from 0.5–2.0%, $n=3$. Illustration created with BioRender.com.

Additionally, we adapted the **Methods** section to state the reason for the exclusion of this group for quantification:

"Characterization of Porosity: ... PEG hydrogels with 0% dextran exhibiting negligible phase separation and featuring sub-micron pores, were omitted from quantification due to the confounding effect of noise affecting the deconvolution of their confocal images, resulting in an overestimation of pore dimensions."

2. The authors report the bulk storage and loss moduli for the gels, but cells interfacing with these materials would not sense the macroscopically determined moduli, but rather the local moduli for the crosslinked gel regions to which they are adherent. Given that osteogenesis by MSCs and osteoblasts is well-known to be regulated by substrate mechanical cues, have the authors characterized the mechanics of the crosslinked phase only (e.g., by micro or nanoindentation)? This may be different than a non-phase separated bulk gel with the same starting PEG concentration, as the authors acknowledge that the phase separation process may result in increased PEG content in the phase separated regions. If mechanical properties vary substantially among conditions, this could prove to be a confounding factor when reading out metrics of osteogenic fate.

Authors' response: We are thankful for this insightful comment. We strongly agree with the reviewer that it will be very interesting to study the local mechanics in these phase-separated hydrogels by micro- and nano-indentation. However, our preliminary AFM experiments were unsuccessful due to the ultra-low stiffness of our hydrogels, which challenge the existing cantilevers for reliable measurements. Nevertheless, how local mechanical cues regulate osteogenic differentiation^[4] have yet to be investigated in future work.

3a. For the rheological characterization of the gels, the authors point out that there is a substantial viscous character to the networks. They speculate that the HA in the system may contribute to the ability of the materials to relax applied stresses. Have the authors tested the stress relaxation rates in these materials? And would this viscoelastic character be expected to remain consistent throughout the culture duration, or can the HA diffuse out of the gels and thereby result in more elastic character over time?

Authors' response: Thank you for your comment. Based on the viscoelasticity demonstrated in current **Revised Supplementary Figure 2d**, we assume that the HA-containing microporous PEG hydrogels, also display stress relaxation properties. We have conducted rheological testing, as shown in the **Revised Figure 2e**, revealing stress relaxation for both degradable and non-degradable PEG hydrogels, respectively. While we anticipate that HA diffusion out of the hydrogel may alter its viscoelasticity towards reduced viscous properties over time, we have not yet investigated the kinetics of this process, which will be explored in future studies. However, we have measured the loss factor (G''/G') after 60

min of crosslinking for PEG hydrogels with and without HA (**Response Figure 3**), showing significantly higher viscosity in hydrogels with HA. Our recent work (Qiu et al. Adv. Funct. Mater. 2023) has demonstrated that adding a viscous gelatin component to synthetic PVA hydrogels renders the matrices stress-relaxing, providing a viscoelastic environment favorable for rapid cell spreading.

Revised Supplementary Figure 2d. Viscoelasticity of microporous PEG hydrogel matrices as determined by frequency sweep measurements on a rheometer at 37 °C (oscillatory strain: 5%, angular frequency: 0.1–100 rad s⁻¹).

Based on the newly acquired stress relaxation data, we have revised **Section 2.2, paragraph 2**, and the corresponding **Revised Figure 2e**:

“Interestingly, a rheological test evidenced the presence of viscoelastic behavior in both degradable and non-degradable PEG hydrogel matrices, likely attributed to the presence of HA (Supplementary Figure 2d). We have recently demonstrated that the presence of a viscous gelatin component in synthetic PVA hydrogels provides stress-relaxation properties and enables rapid cell spreading.³⁷ Our rheological data shows that degradable and non-degradable PEG hydrogels containing HA exhibit stress relaxation at a constant strain of 5% (Figure 2e). This property may facilitate cytoskeletal rearrangements and formation of a 3D cellular network.”

Revised Figure 2e. Stress relaxation of PEG hydrogels measured by rheology at 5% constant strain, $n=3$.

Accordingly, we have updated the **Rheology** section in the **Supplementary Methods** with the addition of the stress relaxation protocol:

“...Stress relaxation tests were performed using a sand-blasted PP20 plate to avoid slipping of the hydrogel during the measurement. Samples of 100 μl hydrogel precursor were crosslinked for 60 min at 37°C surrounded by silicone oil to prevent dehydration while applying 5% oscillatory strain at 1 Hz. Subsequently, the strain was kept constant at 5% for 6 h while monitoring the shear stress. For each sample, the shear stress was normalized to the initial value measured at 5% constant strain. The stress relaxation time $t_{0.6}$ is the time at which the stress reached 60% of the initial value.”

Response Figure 3. Loss factor (G''/G') of PEG hydrogels with and without HA after 60 min of crosslinking showing the effect of HA on viscosity, $n=3$ (**** $p<0.0001$, Student's t-test).

3b. As a minor point, it is not clear from the Figure 2e caption what material composition was tested in the figure panel shown.

Authors' response: The exact hydrogel compositions used in **Revised Supplementary Figure 2d** are indicated in **Supplementary Table 1** which has been updated with the new data and Figure numbering in the revised manuscript. For better clarity, we have included a legend that distinguishes degradable and non-degradable PEG hydrogel in the Figure itself.

4. To demonstrate the osteogenic potential of the embedded cells, the authors report the fraction of area of histological sections that stained positive for mineral deposition and osteocalcin. In these images, it looks like the cell number is also quite different between the degradable and non-degradable materials. Are there differences in cell proliferation between the degradable and non-degradable materials? To account for differences in cell number in the imaged sections, the authors should also present the mineralization and osteocalcin-positive areas normalized by a metric of cell number (e.g., per number of nuclei or per DAPI-positive area). Normalization to cell number will show that the increase in osteogenic signature in the degradable materials is not only due to proliferation differences, but rather to altered ability to spread and form cellular networks.

Authors' response: Thank you for your constructive feedback. As mentioned in the **Supplementary Methods** section, we indeed already normalized the osteocalcin fluorescent signal to the cell count per image (*Osteocalcin Staining of Cryosections*): "To quantify osteocalcin expression, the fluorescence intensity per image was measured as integrated density in Fiji/ImageJ followed by normalization by the cell number in each image." It is noteworthy that the samples were thin cryosections.

Accordingly, we have made this process more explicit in both the figure legend and the caption of revised **Figure 6e** to ensure clarity regarding our normalization approach.

Revised Figure 6e) Quantification of osteocalcin expression by mean fluorescence intensity **normalized to cell number**, $n=3$ (** $p<0.01$, *** $p<0.001$, two-way ANOVA/Tukey).

Due to challenges in counting cells from our light microscopy images, caused by the absence of nuclei counterstaining in these experiments, we were unable to normalize the mineralization signal to cell number.

Nevertheless, we conducted a quantitative DNA assay to elucidate patterns of cell proliferation within our hydrogel matrices. The findings revealed a notable proliferation within degradable and non-degradable hydrogels during the initial 7 days, followed by a decrease in cell numbers across both degradable and non-degradable conditions, as depicted in the **Revised Figure 5b**.

Revised Figure 5b. DNA assay of 3D hMSC culture in degradable and non-degradable PEG hydrogels, $n=3-4$ (* $p<0.05$, ** $p<0.01$, **** $p<0.0001$, two-way ANOVA/Tukey).

Accordingly, we have added a description of this result in the main text (**Section 2.5, paragraph 1**):
 “DNA assay indicated a significant initial increase in cell numbers within both groups over the first 7 days, followed by a reduction in cell numbers (Figure 5b).”

These insights lead us to conclude that while there is an initial surge in cell proliferation within degradable PEG hydrogels, the eventual cell densities align across both hydrogel types after 14 days. The normalization of osteocalcin intensity by cell numbers (**Revised Figure 6e**), alongside comparable cell densities in both hydrogel conditions, supports the conclusion that the improved osteogenic potential observed in degradable materials is not merely due to increased cell proliferation.

5. A major advantage of the gel system cited by the authors is the ability to incorporate the gels into microfluidic organ-on-a-chip systems. The authors show proof of principle that cells encapsulated in these gels can survive physiological levels of shear stress in the microfluidic devices. However, if the authors' goal of a "bone-on-a-chip" is to be realized, the cells need to be able to undergo osteogenic differentiation. Thus, markers of osteogenic differentiation in the dynamic (shear stress) culture conditions should also be assessed.

Authors' response: Thank you for this comment. We conducted a 21-day dynamic culture to address this. Our results show that we are able to achieve flow-enhanced osteogenic differentiation on chip. Please see our changes in the manuscript below.

Based on our promising results on hydrogel permeability using 500 kDa dextran PEG hydrogels (**Figure 7c–e**), we employed an adapted CFD model to estimate the flow rate needed to achieve a physiological FSS of 2 Pa. We adapted **Section 2.6, paragraph 3** accordingly:

“We applied a multiscale and multiphase computational fluid dynamics (CFD) model⁴⁴ to estimate the fluid shear stress (FSS) within the microporous hydrogels (Supplementary Figure 16). Using the

permeability values of PEG hydrogels, a global model on the microfluidic chip was generated to estimate the pressure gradient across the hydrogel region in response to a defined flow rate. This pressure gradient was used for defining the loading conditions of the model on a hydrogel subsection. To determine the flow rate for generating a physiologically relevant average FSS of $\tau_o=2 \text{ Pa}^7$ within the porous space, we conducted a loading (i.e., applied flow rate) variation study using the CFD model based on the pore imaging and permeability data of PEG hydrogels with 500 kDa dextran. Consequently, a flow rate of $Q = 438 \mu\text{l min}^{-1}$ was identified as optimal, producing a pressure drop of 256 Pa across a hydrogel subsection located in the channel center (Figure 7f), resulting in an average FSS of approximately 2 Pa across the hydrogel. Figure 7g depicts FSS distribution in two representative subsections. We used this flow rate to perform an osteogenic on-chip culture for up to 21 days. The estimation of FSS within microporous PEG hydrogels during perfusion serves as a valuable approach to approximate the mechanical cues experienced by embedded cells under dynamic culture conditions. It is important to note that this CFD model does not account for the volumes of embedded cell networks nor cell-secreted nascent proteins on hydrogel permeability. For a deeper understanding, future studies are required by considering the dynamics of cell-matrix remodeling.”

The **Supplementary Methods** section **Computational Fluid Dynamics (CFD)** was adapted accordingly:

“*Computational Fluid Dynamics (CFD) Simulation:* Confocal images of rhodamine-labeled PEG hydrogel on chip (AIM Biotech, DAX-1) were processed, and the pore geometry was reconstructed using Seg3D (University of Utah, UT, USA). To quantify the FSS within the scaffold that has highly irregular porous geometries, a multiscale and multiphase CFD model previously developed⁴⁴ was used. The model involves 2 scales, i.e. (i) the global scale that represents the whole microfluidic chip and (ii) the local scale that models the detailed micro-structures of subsections (dimension: $20 \times 20 \times 30 \mu\text{m}$, $n=4$) from the whole scaffold. In the global model, the scaffold region was modelled as porous media with a permeability of $8.67 \times 10^{-15} \text{ m}^2$, which was obtained from experimental measurement of a hydrogel with a composition containing 2.2% 4-PEG-VS and 1.0% 40 kDa dextran matching the confocal microscopy data. To approximate the experimental condition for dynamic cell culture, two types of flow rates (i.e., $10 \mu\text{l min}^{-1}$ and $100 \mu\text{l min}^{-1}$ per port) were applied to the global model as inlet and outlet boundary conditions. Mass flux conservation was applied to the interface between porous media and free fluid. The global model was meshed with 450410 tetrahedral elements. The pressure gradient that was calculated from the global model was applied to the local CFD model for simulating the shear stress on PEG scaffold surfaces. The boundary conditions of local CFD model are shown in **Supplementary Figure 16**. In the local model, the fluid domain of each subsection was meshed by a uniform tetrahedral element size of $0.4 \mu\text{m}$, which generated 1322072, 1135226, 1247121, and 1302269 elements, respectively, for 4 discretized subsections. In this study, the fluid was modelled as laminar flow with the dynamic viscosity of DMEM ($7.8 \times 10^{-4} \text{ Pa s}$). The CFD models were solved by a finite volume method (FVM) using ANSYS CFX (ANSYS Inc., PA, USA) under the convergence criteria of root-mean-square residual of the mass and momentum $<10^{-4}$. The same methods for (i) confocal images – based geometry reconstruction and (ii) CFD model setup were applied to the PEG hydrogel scaffold with 500 kDa dextran with a permeability of $2.97 \times 10^{-13} \text{ m}^2$. In the local model, 4 subsections were meshed with 1664388, 1790433, 1398725 and 1449911 elements, respectively. The varying flow rates of 100 – 1000

$\mu\text{L}/\text{min}$ were applied to the inlets of the chip (global model), under which the corresponding FSS within each subsection was computed. Then the optimal flow rate was recorded for the desirable average FSS.”

In addition, we adapted the corresponding **Revised Figure 7** with this newly acquired data and moved the previously acquired CFD data in the Supplementary Information as **Revised Supplementary Figure 17**:

Revised Figure 7. Integration of PEG hydrogels with a microfluidic chip for perfusion of the microporous matrix. a) Microfluidic setup to visualize fluid flow through PEG hydrogels, Q_1 and Q_2 denote volumetric flow rate, P_1 and P_2 represent positions of image acquisition. b) Time-lapsed fluorescence microscopy images of FITC-dextran (500 kDa) tracer perfusing through PEG hydrogels on chip in response to a pressure gradient, scale bar: 200 μm . c) Microfluidic setup for permeability test depending on a gravity-driven fluid flow through the PEG hydrogels, h_1 and h_2 denote measured heights, h_d is the height difference to calculate the changes in pressure drop over time. d) Calculated pressure change over time (symbols) and fitted exponential functions (lines) across 40 kDa and 500 kDa dextran PEG hydrogels and reference hydrogels made of 2 mg ml^{-1} collagen type I, $n=3$. e) Resulting Darcy’s permeability of the 3 hydrogels obtained from pressure change over time. f) Pressure distribution within the porous media (homogenized scaffold domain) under the applied flow rate of 438 $\mu\text{l min}^{-1}$ per inlet. h) Fluid shear stress (FSS) distribution and average FSS (τ_a) within two representative subsections (x-y-z: 20 x 20 x 30 μm) under an applied flow rate of 438 $\mu\text{l min}^{-1}$ per inlet. Illustrations in a) and c) created with BioRender.com.

Revised Supplementary Figure 17. Computational fluid dynamics model to estimate the fluid shear stress (FSS) distribution within PEG hydrogels with 40 kDa dextran. a) Pressure distribution within the porous media (homogenized scaffold domain) under the applied flow rate of $10 \mu\text{l min}^{-1}$ per inlet. b) FSS distribution and average FSS (τ_a) within a representative subsection (x-y-z: $20 \times 20 \times 30 \mu\text{m}$) under an applied flow rate of $10 \mu\text{l min}^{-1}$ per inlet.

Based on the results of this simulation, we conducted an additional dynamic osteogenic cell culture on chip using hMSC. **Section 2.7, paragraph 3-4** was adapted accordingly:

“Given their **microporosity**, permeability and permissiveness for differentiation, we lastly examined the potential of PEG hydrogels for dynamic cell culture on a chip. **Initially**, we studied the effect of controlled delivery of FSS on the viability of **hMSC embedded in microporous hydrogels with 40 kDa dextran**. Cells were subjected to perfusion using either a low flow rate ($Q_{\text{low}}=10 \mu\text{l min}^{-1}$ per inlet) or high flow rate ($Q_{\text{high}}=100 \mu\text{l min}^{-1}$ per inlet), resulting in an average FSS of $\tau_a=1.74$ Pa and $\tau_a=17.43$ Pa, respectively, according to CFD simulations (Supplementary Figure 17). Live/dead assay results on day 13 revealed comparable cell viability between static and low FSS culture, while high FSS led to a significant reduction in cell viability (Supplementary Figure 18), likely attributed to the high shear stresses well beyond the physiological range within the LCN porosities in vivo ($0.8\text{--}3.0$ Pa).⁷

Building on these findings, we used microporous PEG hydrogels for a 21-day bone-on-chip perfusion culture. For this, we utilized PEG hydrogels with higher permeability (500 kDa dextran) and the estimated flow rate from the CFD model to apply a physiological average FSS of $\tau_a \approx 2$ Pa (Figure 7g) for embedded hMSC. Perfusion was applied 3 times per week for 10 min each (dynamic, Figure 8c) and lasted for 3 weeks with static on-chip cultures as controls. Live-dead staining (Figure 8d, Supplementary Figure 19) showed high cell viability in both conditions (static and dynamic) on day 7 and 21 of culture. No significant differences were observed between the groups over time. Immunofluorescence staining for the osteoblast marker osteocalcin3 revealed similar expression levels on day 21 for both conditions, but a significant increase from day 7 to day 21 for dynamic cultures (Supplementary Figure 20a, b).

Podoplanin, as a marker for dendrite formation of embedding osteocytes⁴¹, was observed on day 7 in dynamic cultures but not in static ones. Although expression levels were higher in static conditions on day 21, podoplanin expression in dynamic cultures appeared more confined within dendrites. FSS has not only been shown to enhance *in vitro* osteogenic differentiation but also to induce matrix mineralization.^{7, 45} Here, we assessed mineral formation in both static and dynamic hMSC cultures within PEG hydrogels after 21 days (Figure 8e). Alizarin red staining showed significantly enhanced mineral deposition in the dynamic over static cultures. Together, this proof-of-concept experiment highlights the potential of microporous PEG hydrogels for dynamic cell culture on a chip.”

Please find the corresponding changes of Figures and Methods Section below:

Supplementary Methods, hMSC Culture:

“...For the preliminary dynamic cell culture, fluid shear stress (FSS) was applied on chip (AIM Biotech, DAX-1) by connecting two syringe pumps to both inlets of one medium channel and applying a total flow rate of $20 \mu\text{l min}^{-1}$ (low FSS) or $200 \mu\text{l min}^{-1}$ (high FSS) using DMEM. Loading was performed 2×10 min daily starting on day 3 until day 7 of culture and then again from day 10 until day 13 with 60 min in between each treatment. Three replicates were used per condition (static, low FSS, high FSS). An additional dynamic cell culture was performed using a 24-channel peristaltic pump (Longer, BT100-1L) to apply a flow rate of $438 \mu\text{l min}^{-1}$ (corresponding to a FSS $\tau_a=2$ Pa, as simulated by the CFD model) to dynamic samples. Perfusion was performed using DMEM supplemented with 1% Anti-Anti for 10 min 3 times per week for up to 21 days. Static controls were cultured on chip, and osteogenic medium was exchanged 3 times per week.”

Supplementary Methods, Alizarin Red Staining:

“To assess matrix mineralization in cryosections and within microfluidic chips, calcium deposits were stained with Alizarin red. A staining solution (2 mg ml^{-1} Alizarin red S (Sigma-Aldrich, A5533-25G) in distilled water, pH adjusted to 4.12) was applied to the samples after two washes with distilled water. The samples were stained for 30 min at room temperature and then washed 5 times until the water came out clear. Imaging at $5 \times / 10 \times / 20 \times$ magnification was performed on Leica DMI1 microscope. A color threshold was applied by selecting only the red channel in RGB color space in Fiji/ImageJ. The area of red color was then measured for each condition.”

Figure 8. 3D microfluidic culture of hMSC within PEG hydrogels. **a)** Phase contrast (scale bars: 200 μm) and confocal microscopy images (scale bars: 100 μm) showing cell network formation of hMSC embedded in degradable and non-degradable PEG hydrogels on chip after 2 days of osteogenic culture. **b)** Phase contrast and confocal microscopy images showing the effect of high ($1 \times 10^6 \text{ ml}^{-1}$) and low ($5 \times 10^5 \text{ ml}^{-1}$) cell seeding density on hMSC morphology inside MMP-degradable PEG hydrogels on day 7 of osteogenic culture on chip, scale bars: top - 500 μm , middle - 200 μm , bottom - 50 μm . **c-e)** 21-day hMSC culture within degradable PEG hydrogels on chip under static and dynamic (FSS $\tau_a \approx 2 \text{ Pa}$) conditions. **c)** Schematic illustration of applied FSS to microfluidic hMSC culture ($Q=438 \mu\text{l min}^{-1}$ per inlet). **d)** Cell viability on day 7 and 21 of hMSC embedded in MMP-degradable PEG hydrogels on chip in response to FSS or static culture, $n=3$ (two-way ANOVA/Tukey). **e)** Matrix mineralization after 21 days in osteogenic culture determined by Alizarin red staining. Left: microscopy images, scale bars: 200 μm ; right: quantification of stained area fraction, $n=3$ (** $p < 0.001$, Student's t-test). Illustrations in a) and c) created with BioRender.com.

Supplementary Figure 18. Preliminary dynamic microfluidic culture of hMSC within PEG hydrogels (40 kDa dextran). **a)** Schematic representation of experimental setup to apply fluid shear stress (FSS) to cells embedded in PEG hydrogels on chip. Illustration created with BioRender.com. **b)** Confocal microscopy images (MIPs) of live-dead staining, scale bars: 200 μm . **c)** Quantification of cell viability based on live/dead staining, $n=3$ (* $p < 0.05$, one-way ANOVA/Tukey).

Supplementary Figure 19. Confocal microscopy images (MIPs) of live-dead stained hMSC after 7 and 21 days of static and dynamic osteogenic culture within degradable PEG hydrogels, scale bars: 200 μm .

Supplementary Figure 20. 21-day hMSC culture within degradable PEG hydrogels on chip with and without the application of FSS $\tau_a \approx 2$ Pa as simulated using the CFD model. **a)** Confocal microscopy images (MIPs) of osteocalcin and podoplanin immunofluorescence staining as osteoblast and early osteocyte markers, respectively, scale bars: 100 μm . **b)** Quantification of fluorescence intensity of immunostaining normalized to cell number for osteocalcin and podoplanin, $n=3$ (* $p<0.05$, ** $p<0.01$, two-way ANOVA/Tukey).

6. The authors' assertion that the PIPS method uniquely allows for pore formation in the presence of live cells is not accurate based on the papers cited to support this statement (lines 279-281). For instance, ref. 16 has pores form while live cells are present due to hydrolytic degradation of a sacrificial porogen phase. The system in the present manuscript does have the advantage of requiring only a single processing step (phase separation during crosslinking) in order to generate the porous structures.

Authors' response: Thank you for pointing this out. We have adapted this section based on your suggestions (**Section 1, paragraph 2**):

“Various techniques, such as emulsification¹⁷, porogen leaching¹⁸⁻²¹ and particle or microgel annealing²²⁻²⁶ have been employed to create microporous hydrogels with relevant pore sizes for cell spreading of 5-150 μm . Nevertheless, most of these methods have limitations in generating interconnected pores in the presence of living cells, while others require the degradation of a sacrificial porogen phase and therefore multiple processing steps.”

Reviewer #3 (Remarks to the Author):

The manuscript reports the engineering of MMP-sensitive PEG hydrogels for supporting bone formation. The work presents an interesting study design but lacks the novelty and impact in respect to performance of the proposed hydrogels. Moreover, it is not clear how the synthetic hydrogel performs as compared to the gold standard material. Several important techniques are missing in order to better characterise the hydrogels in respect to bioactivity and osteogenic ability.

Authors' response: We are grateful for the reviewer's feedback and constructive suggestions about additional characterization of our hydrogels for osteogenic differentiation.

1. The Introduction section should be improved to better reflect the current state of the art.

Authors' response: Thank you for this comment. In response to Comment 2 from Reviewer #1, we have added additional relevant literature to the Introduction section to better reflect the state-of-the-arts in developing microporous and granular hydrogels for functional tissue engineering.

2. Figure 1 schematics are not scientifically accurate and is too oversimplistic.

Authors' response: Thank you for your feedback to Figure 1. We admit that the relative length scales of visuals such as hydrogel networks and pores may not match the ratio in reality. We chose a simplified illustration to convey the principle of our void-forming hydrogels with pore sizes (5-20 μm) and how they uniquely facilitate rapid 3D cell network formation to address a broad audience, including researchers who may not be experts in the field.

We have also adapted the figure by changing the heading "Void-forming hydrogels" to "This work" (see below). This adjustment aims to clarify that the schematic highlights the unique contributions of our research.

We have revised the main text as follows to highlight the importance of appropriate pore size for engineering 3D cell networks (Section 1, paragraph 2):

“...However, these hydrogels often have nanoscale pore sizes (5–100 nm) and limited permeability.¹² Consequently, cell spreading relies on matrix degradation via hydrolysis or proteolysis through cell-secreted matrix metalloproteinases (MMPs). In contrast, top seeding typically relies on scaffolds with large pores (100–600 μm)^{13–16} where the cell to surface interface is 2D. As a result, cells often fail to form 3D networks. Various techniques, such as emulsification¹⁷, porogen leaching^{18–21} and particle or microgel annealing^{22–26} have been employed to create microporous hydrogels with relevant pore sizes for cell spreading of 5–150 μm .”

Figure 1. Schematic illustration of conventional biomaterials versus void-forming polyethylene glycol (PEG) hydrogels for bone tissue engineering. Traditional nanoporous hydrogels (pore sizes: 5–100 nm¹²) impede cell spreading, while macroporous scaffolds with large pores (pore sizes: 100–600 μm ^{13–16}) merely provide a 2D cell surface interface. Herein, **microporous** PEG hydrogels (pore sizes: 5–20 μm) are developed for rapid 3D cell network formation. Illustration created with BioRender.com.

3. The degradation of the synthetic hydrogels should be carried out at different pH's.

Authors' response: We like to point out that the main mechanism of hydrogel degradation is through proteolysis induced by cell-secreted matrix metalloproteinases as described in our previous work (Qin et al. Adv. Mater. 2018).^[2] Therefore, testing hydrogel degradation at different pH levels will not make sense given the cell-dependent nature of degradation and the needs for precise pH control in cell culture.

4. The cell viability and proliferation should be performed until longer culture times (e.g. 14 days). Different osteogenic markers should be quantitatively investigated at early and late stage by RT PCR, and gold standard materials should be used as controls (e.g. hydroxyapatite).

Authors' response: Thank you for this feedback. We have conducted an additional 21-day static cell culture experiment using hMSC to investigate long-term cell viability, proliferation and quantitative markers of osteogenic differentiation. To our knowledge, the gold standard material to model early bone development *in vitro* (prior to extensive mineralization) is collagen type I hydrogel. This material has been extensively used in literature, e.g. by Nasello et al.^[1], showing promising results in terms of osteogenic ability. Our newly acquired data suggests that our PEG hydrogels perform similarly well in terms of osteogenic differentiation with the advantage of having a synthetic matrix that allows for visualization of cell-secreted ECM. Please find our corresponding changes in the manuscript below.

We added an additional paragraph and **Figure 5** to **Section 2.5**:

“2.5. Osteogenic Differentiation of hMSC in Static PEG Hydrogel Cultures

Next, we investigated the osteogenic differentiation of hMSC in microporous PEG hydrogels as a function of matrix degradability. A live-dead staining assay revealed high cell viability (>95%) in both types of hydrogels at 14 and 21 days (Figure 5a, Supplementary Figure 11). A DNA assay indicated a significant initial increase in cell numbers within both groups over the first 7 days, followed by a reduction in cell numbers (Figure 5b). Alkaline phosphatase (ALP) activity was significantly higher in degradable hydrogels at 14 and 21 days (Figure 5c). Furthermore, real-time quantitative PCR (RT-qPCR) data showed a pronounced increase in *ALPL* gene expression in the degradable hydrogels from day 7 to day 21, compared to the non-degradable ones (Figure 5d). *RUNX2* gene expression was higher in degradable hydrogels at 14 days (Figure 5e). Additionally, *MMP14* and *PDPN*, involved in osteocytic dendrite formation⁴¹, were expressed to a greater extent in degradable hydrogels (Figure 5f, g). Immunofluorescence staining revealed expression of collagen type I in both hydrogels over time (Supplementary Figure 12a, c; negative controls in Supplementary Figure 13). However, collagen I appeared more widely distributed across the degradable hydrogels compared to the non-degradable ones. Notably, the expression of podoplanin was significantly higher in degradable PEG hydrogels than that in non-degradable ones at day 21 (Supplementary Figure 12b, d).”

Figure 5. Quantitative analysis of static 21-day osteogenic hMSC culture within MMP-degradable and non-degradable PEG hydrogels. **a)** Cell viability of hMSC after 14 and 21 days of osteogenic culture based on live/dead staining, $n=3$ (two-way ANOVA/Tukey). **b)** DNA assay of 3D hMSC culture in degradable and non-degradable PEG hydrogels, $n=3-4$ ($*p<0.05$, $**p<0.01$, $****p<0.0001$, two-way ANOVA/Tukey). **c)** ALP activity (measured as p-nitrophenol concentration) of 3D hMSC culture in degradable and non-degradable PEG hydrogels normalized to DNA content, $n=3-4$ ($*p<0.05$, $**p<0.01$, $***p<0.001$, two-way ANOVA/Tukey). **d-g)** Quantification of gene expression levels using qPCR for markers of osteoblasts (*ALPL*, *RUNX2*), osteoblast-to-osteocyte transition (*MMP14*), and osteocytes (*PDPN*) over 3 weeks normalized to *ACTB* expression, $n=3$ ($*p<0.05$, $**p<0.01$, $***p<0.001$, $****p<0.0001$, two-way ANOVA/Tukey).

Supplementary Figure 11. Cell viability of hMSC after 14 and 21 days of osteogenic culture within degradable and non-degradable PEG hydrogels. Confocal microscopy images (MIPs) of live/dead staining, scale bars: 200 μm .

Supplementary Figure 12. Immunofluorescence staining of static 21-day hMSC culture in degradable and non-degradable PEG hydrogels. **a–b)** Confocal microscopy images (MIPs) of collagen I and early osteocytic marker podoplanin, scale bars: 100 μ m. **c–d)** Quantification of fluorescence intensity of immunostaining normalized to cell number for collagen I and podoplanin within degradable and non-degradable PEG hydrogels, $n=3$ (* $p<0.05$, ** $p<0.01$, two-way ANOVA/Tukey).

Supplementary Figure 13. Control experiments for immunofluorescence staining omitting the primary antibody, utilizing only the secondary antibody-conjugates with AF488 or AF647 for staining, scale bars: 100 μm . The samples tested are hMSC cultured for 21 days in degradable PEG hydrogels.

The methods related to this new data can now be found in the **Supplementary Methods** section:

Alkaline phosphatase (ALP) assay: The enzymatic activity of ALP in cells within PEG hydrogels was assessed using a colorimetric assay based on the enzyme's ability to convert p-nitrophenylphosphate into p-nitrophenol. PEG hydrogels containing 10^5 cells each were washed with PBS. Subsequently, 0.5 ml of 0.2% w/v Triton X-100 with 5 mM MgCl_2 was added to each hydrogel in an Eppendorf tube, followed by a 30-minute incubation period. Hydrogels were then homogenized using Fisherbrand™ Pellet Pestle™, and the homogenized samples were centrifuged at 3000 g for 10 min. Following centrifugation, 200 μl of the supernatant was collected in new Eppendorf tubes. For the ALP assay, the following reagents were pipetted into a clear flat-bottom 96-well plate: 80 μl of the sample or standard, 20 μl of 0.75 M 2-amino-2-methyl-1-propanol (AMP, pH adjusted to 10.5, Sigma, A65182) and 100 μl of substrate solution. The substrate solution was prepared by mixing 37.11 mg of p-nitrophenylphosphate disodium salt hexahydrate (Sigma, 71768-5G) with 1 ml of 1.5 M AMP buffer and 9 ml of ultrapure water (UPW). P-nitrophenol standards (Sigma-Aldrich, 425753) were prepared from a 1 mM stock solution in 0.2% w/v Triton X-100 with 5 mM MgCl_2 and diluted to concentrations of 0, 0.05, 0.2, 0.6 and 0.9 $\mu\text{mol ml}^{-1}$. The reaction was left to proceed for 15 min before 100 μl 0.2 M NaOH was added to stop the reaction. The absorbance at 405 nm was then measured on a plate reader (Tecan, Spark 10M). Samples and standards were measured in technical duplicate and averaged, and the absorbance of the zero standard was subtracted from all samples and standards before the concentration of p-nitrophenol was calculated from the standard curve. The obtained concentrations were normalized by the reaction time (15 min) and finally by the DNA content in each sample as detailed below.

DNA assay: After completing the ALP assay, the remaining volume of homogenized hydrogels underwent three cycles of freeze-thawing and ultrasonication. Subsequently, the samples were incubated at room temperature for 48 hours. DNA quantification was performed using the Quant-iT PicoGreen dsDNA assay kit (Invitrogen, P7589) according to the manufacturer's instructions. In short, 87.5 μl of 1 \times TE buffer, 12.5 μl of the sample and 100 μl of PicoGreen working solution were added to each well of a clear, flat-bottom 96-well plate. Standards were prepared according to the manufacturer's protocol. Samples and standards were measured in duplicate and the results averaged. After 5 min of incubation in the dark, fluorescence emission at 535 nm with an excitation at 485 nm was measured using the plate reader. For quantification, the zero standard was subtracted from all measurements and the sample DNA concentration was calculated from the standard curve.

RNA Isolation and Reverse Transcription: PEG hydrogels with hMSC ($\sim 10^5$ cells per sample) were cultivated for 0, 7, 14 or 21 days in osteogenic medium, washed with PBS and shock frozen in liquid N_2 . Samples were stored at $-80^\circ C$ until further use. Hydrogels were thawed and homogenized in 300 μl Trizol (Invitrogen, 15596018) with a Fisherbrand™ Pellet Pestle™ and a 21G needle. 700 μl Trizol were added to each sample and samples were incubated for 5 min at room temperature. Samples were centrifuged for 30 s at 8500 g and the supernatant was collected. 200 μl Chloroform (Sigma-Aldrich, C2432-500ML) were added before vortexing. After incubating for 5 min, samples were centrifuged for 20 min at 12000 g at $4^\circ C$. The aqueous phase was collected and 1 volume of 70% EtOH was added. The samples were then transferred on RNeasy MinElute spin columns and the manufacturer's protocol of the RNeasy Micro kit (Qiagen, 74004) was followed with a few exceptions: Washing steps 4, 6 and 7 were repeated twice each and RNA was eluted in 28 μl H_2O . 20–22 μl RNA were reversed transcribed using iScript™ Reverse Transcription Supermix for RT-qPCR (Bio-Rad, 1708840) according to the manufacturer's protocol. After synthesis, the cDNA was diluted 1:5 in ultrapure H_2O .

Quantitative PCR (qPCR): 7.2 μl of cDNA were mixed with 8 μl TaqMan™ Fast Universal PCR Master Mix (2X), no AmpErase™ UNG (Applied Biosystems, 4366072) and 0.8 μl TaqMan™ Gene Expression Assay Primers (Supplementary Table 2). A dilution series of 1v:1v (1, 1/2, 1/4, 1/8, 1/16) of a standard sample was used for relative quantification of the samples. Samples were initially heated 30 s at $95^\circ C$ followed by up to 50 cycles of 5 s at $95^\circ C$ and 20 s at $60^\circ C$ in a CFX96™ Real-Time System C1000 Touch™ Thermal Cycler. Samples were analyzed in technical duplicate. Gene expression data was normalized to the housekeeping gene β -actin (*ACTB*).

Supplementary Table 1. List of primers used for qPCR.

Gene	TaqMan ID
Human ACTB	Hs01060665_g1
Human PDPN	Hs00366766_m1
Human RUNX2	Hs00231692_m1
Human MMP14	Hs00237119_m1
Human ALPL	Hs01029144_m1

Immunofluorescence Staining of 3D Hydrogels: To investigate the expression of osteogenic markers in 3D hydrogels in molds and on chip, fixed hydrogels were first incubated in 0.3% w/v Triton X-100 in PBS for 20 min. Non-specific antibody binding was then blocked with 1% BSA w/v and 5% serum v/v from the host of the secondary antibody for 45 min (donkey serum, Abcam, ab7475). Primary antibodies, including Anti-Osteocalcin (rabbit, Abcam, ab93876), Anti-Collagen I (mouse, Abcam, ab6308), and Anti-Podoplanin (mouse, Santa Cruz Biotechnology, sc-59347), were diluted to a 1:200 concentration in the same blocking buffer. Samples were then incubated overnight at $4^\circ C$ in this solution. To serve as a negative control, primary antibodies were omitted in selected samples. Following primary antibody incubation, samples were washed 3×5 min in 0.025% w/v Triton X-100 in PBS. Corresponding secondary antibodies (donkey anti-mouse IgG AF488, A21202 and donkey anti-rabbit IgG AF 647, A-31573) were diluted to 1:500 in 1% BSA in 0.3% Triton X-100 in PBS each. To counterstain the cytoskeleton and nuclei, phalloidin-TRITC and Hoechst 33342, respectively, were added at a dilution of 1:500. Samples were incubated in the secondary antibody solution for 2 h protected from light before washing them 3×5 min in 0.025% w/v Triton X-100 in PBS. 100 μm thick z-stacks were acquired using a 20 \times air objective on a Zeiss LSM 780 confocal microscope. To quantify osteogenic marker expression, the fluorescence intensity per image was measured as integrated density in Fiji/ImageJ followed by normalization by the cell number in each image."

6. Mineralization and osteogenic markers should be quantitatively investigated in the 3D microfluidic culture of cell-laden PEG hydrogels.

Authors' response: Thank you for this feedback. Please read our new results in response to this comment and also to comment 5 from **Reviewer #2**.

References

1. G. Nasello, P. Alamán-Díez, J. Schiavi, M.Á. Pérez, L. McNamara, and J.M. García-Aznar, *Primary human osteoblasts cultured in a 3D microenvironment create a unique representative model of their differentiation into osteocytes*. *Frontiers in bioengineering and biotechnology*, 2020. **8**: p. 336.
2. X.H. Qin, X. Wang, M. Rottmar, B.J. Nelson, and K. Maniura-Weber, *Near-infrared light-sensitive polyvinyl alcohol hydrogel photoresist for spatiotemporal control of cell-instructive 3D microenvironments*. *Advanced Materials*, 2018. **30**(10): p. 1705564.
3. M. Lutolf and J. Hubbell, *Synthesis and physicochemical characterization of end-linked poly(ethylene glycol)-co-peptide hydrogels formed by Michael-type addition*. *Biomacromolecules*, 2003. **4**(3): p. 713-722.
4. A. Cipitria, C. Lange, H. Schell, W. Wagermaier, J.C. Reichert, D.W. Hutmacher, P. Fratzl, and G.N. Duda, *Porous scaffold architecture guides tissue formation*. *Journal of bone and mineral research*, 2012. **27**(6): p. 1275-1288.

Reviewers' Comments:

Reviewer #1:

Remarks to the Author:

The authors have already addressed my previous comments and made significant improvements to the manuscript's quality. Yet, there are a few issues or questions should be addressed.

1. In Figure 7f, the applied flow rate is $438 \mu\text{l min}^{-1}$ per inlet. While the storage modulus of microporous PEG hydrogels may be lower than 200 Pa or even lower than 150 Pa for different concentration configurations according to Figure 2, does this flow rate ($438 \mu\text{l min}^{-1}$) cause deformation in the microporous hydrogels? Or would there be the stress relaxation in the viscoelastic microporous hydrogels on chip during or after each perfusion cycle that exerts pressure on the hydrogel? Should these considerations also be included in the CFD model? Perhaps discussing these aspects would benefit interdisciplinary readers.
2. The perfusion on chip was applied 3 times per week for 10 min each and lasted for 3 weeks. Can the 10-minute perfusion provide enough stimulation of fluid shear stress on cells during each application? It is possible that cells may require some time to respond upon receiving external stimuli.
3. "The microfluidic setup (Figure 7c) was employed to quantify hydrogel permeability". According to equation 2, the calculation for Permeability K would involve the constant c and A , where A represents the cross-sectional area of the hydrogel channel in the direction of fluid flow. Could authors provide the specific value for cross-sectional area A ?"
4. If the AFM measurement was not suitable for the ultra-low stiffness of the hydrogels in this study, is there some possibilities to use other methods based on Brillouin light-scattering or encapsulated magnetic microbeads in the hydrogel to investigate how local mechanical cues regulate the cellular behaviors? It would be better if authors could provide some discussion on these possibilities. It would enhance the quality of the paper and benefit readers.

Reviewer #2:

Remarks to the Author:

The authors have sufficiently addressed the reviewers' concerns.

Reviewer #3:

Remarks to the Author:

The authors have addressed the reviewer major comments in a proper manner. The manuscript should be accepted for publication.

REVIEWER COMMENTS

Reviewer #1 (Remarks to the Author):

The authors have already addressed my previous comments and made significant improvements to the manuscript's quality. Yet, there are a few issues or questions should be addressed.

Authors' response: We appreciate the reviewer's positive feedback on the revised manuscript, as well as the constructive suggestions offered to improve it further.

1. In Figure 7f, the applied flow rate is 438 $\mu\text{l min}^{-1}$ per inlet. While the storage modulus of microporous PEG hydrogels may be lower than 200 Pa or even lower than 150 Pa for different concentration configurations according to Figure 2, does this flow rate (438 $\mu\text{l min}^{-1}$) cause deformation in the microporous hydrogels? Or would there be the stress relaxation in the viscoelastic microporous hydrogels on chip during or after each perfusion cycle that exerts pressure on the hydrogel? Should these considerations also be included in the CFD model? Perhaps discussing these aspects would benefit interdisciplinary readers.

Authors' response: Thanks for your insightful comments. While we have not observed significant deformations of the hydrogel at lower flow rates, applying a flow rate of 438 $\mu\text{l min}^{-1}$ during perfusion may lead to deformation of the microporous hydrogels. This could result in mechanical stress to embedded cells. In our study, perfusion was only applied for 10 minutes each time (3 times per week), which should allow for sufficient stress relaxation in the hydrogel. Additionally, we expect the gradual stiffening of the hydrogel due to the secretion of ECM proteins over time. It has been shown that strain-induced fluid flow plays a more significant role in osteoblast mechanosensation than matrix strain, even in collagen matrices stiffened with hydroxyapatite.^[1]

To capture the deformation of hydrogel struts under perfusion, a fluid-structure interaction (FSI) model has yet to be employed. However, the quantitative evolution of mechanical properties of the cell-hydrogel struts are still unknown for calculating the deformation of hydrogel struts by FSI model. This will be extremely interesting to explore in our future work.

Accordingly, we have adapted the discussion of the CFD model (**Section 2.6, paragraph 3**):

“Additionally, hydrogel deformation under perfusion is not considered. For a deeper understanding, future studies should unravel the dynamics of cell-matrix interplay and flow-induced hydrogel deformation using a fluid-structure interaction model.”

2. The perfusion on chip was applied 3 times per week for 10 min each and lasted for 3 weeks. Can the 10-minute perfusion provide enough stimulation of fluid shear stress on cells during each application? It is possible that cells may require some time to respond upon receiving external stimuli.

Authors' response: Thank you for this question. Our perfusion regimen, consisting of 10-minute perfusion sessions (three times per week for three weeks), was inspired by the seminal work by Robling et al. **“Shorter, more frequent mechanical loading sessions enhance bone mass”**^[2] as well as recent *in vitro* studies employing cyclic compressive loading.^[3-5] Some of these studies utilized a similar regimen of 5 minutes of compressive loading (three to five times a week) on 3D bioprinted bone constructs, which resulted in enhanced osteogenic differentiation.

Numerous *in vitro* studies have demonstrated that both osteoblasts and osteocytes rapidly respond to fluid shear stress (FSS) as mechanical stimulus, triggering Ca^{2+} signaling and downstream pathways.^[6]

Using live-cell Ca^{2+} imaging, we recently demonstrated that mouse IDG-SW3 osteocytes rapidly exhibited Ca^{2+} spikes in response to perfusion stimulation on chip, indicating transient cellular response to FSS.^[7]

In this study, we demonstrated proof-of-concept perfusion culture using the above-mentioned regimen on an AIM BIOTECH chip to promote osteogenic differentiation of embedded hMSC in microporous hydrogels. [Redacted]

3. "The microfluidic setup (Figure 7c) was employed to quantify hydrogel permeability". According to equation 2, the calculation for Permeability K would involve the constant c and A, where A represents the cross-sectional area of the hydrogel channel in the direction of fluid flow. Could authors provide the specific value for cross-sectional area A?"

Authors' response: Thank you for your valuable comment. We have updated the specific values for the cross-sectional area of the hydrogel channel (A) and the syringe barrel (A_r) in the **Supplementary Methods, Quantification of Permeability:**

"...Permeability K and the constant c are related according to Darcy's law in **Equation 2**, where μ is the viscosity of DMEM (7.8×10^{-4} Pa s), l is the length of the hydrogel channel (1.70×10^{-2} m), A_r is the cross section of the syringe barrel (**9.62 mm²**) and A is the cross section of the hydrogel channel in the direction of fluid flow (**1.52 mm²**)."

Additionally, during the revision process, we identified an error in the unit used for the exponent coefficient (c) in **Figure 7e**. The correct unit should be h^{-1} , not s^{-1} as previously stated in the Table. This correction does not affect the permeability results.

Revised Figure 7. Integration of PEG hydrogels with a microfluidic chip for perfusion of the microporous matrix. **a)** Microfluidic setup to visualize fluid flow through PEG hydrogels, Q_1 and Q_2 denote volumetric flow rate, P_1 and P_2 represent positions of image acquisition. **b)** Time-lapsed fluorescence microscopy images of FITC-dextran (500 kDa) tracer perfusing through PEG hydrogels on chip in response to a pressure gradient, scale bar: 200 μm . **c)** Microfluidic setup for permeability test depending on a gravity-driven fluid flow through the PEG hydrogels, h_1 and h_2 denote measured heights, h_d is the height difference to calculate the changes in pressure drop over time. **d)** Calculated pressure change over time (symbols) and fitted exponential functions (lines) across 40 kDa and 500 kDa dextran PEG hydrogels and reference hydrogels made of 2 mg ml⁻¹ collagen type I, $n=3$. **e)** Resulting Darcy's permeability of the 3 hydrogels obtained from pressure change over time. **f)** Pressure distribution within the porous media (homogenized scaffold domain) under the applied flow rate of 438 μl min⁻¹ per inlet. **g)** Fluid shear stress (FSS) distribution and average FSS (τ_a) within two representative subsections (x-y-z: 20 x 20 x 30 μm) under an applied flow rate of 438 μl min⁻¹ per inlet. Illustrations created with BioRender.com.

4. If the AFM measurement was not suitable for the ultra-low stiffness of the hydrogels in this study, is there some possibilities to use other methods based on Brillouin light-scattering or encapsulated magnetic microbeads in the hydrogel to investigate how local mechanical cues regulate the cellular behaviors? It would be better if authors could provide some discussion on these possibilities. It would enhance the quality of the paper and benefit readers.

Authors' response: Thank you for this valuable suggestion to explore alternative methods for studying the effect of local mechanical cues on embedded cells. We agree that methods based on

Brillouin light-scattering or encapsulated microbeads will be suitable to measure the local mechanical properties of our hydrogels. However, these methods have not yet been established in our lab.

In response to your suggestion, we have extended our discussion on the mechanical properties of the hydrogels, highlighting the limitation of not having tested the local mechanical properties and discussing potential methods for investigating this in the future (**Section 2.2, paragraph 3**):

“... It is important to note that rheological analysis on the bulk hydrogels may not reflect the local hydrogel mechanics experienced by cells. Due to the low G' , atomic force microscopy-based nanoindentation was unsuccessful. In the future, microrheology using nanosized beads may help dissect how mechanical heterogeneity in microporous hydrogels impacts cell behaviors³⁸”

References

1. S.M. Tanaka, H.B. Sun, R.K. Roeder, D.B. Burr, C.H. Turner, and H. Yokota, *Osteoblast responses one hour after load-induced fluid flow in a three-dimensional porous matrix*. *Calcified tissue international*, 2005. **76**: p. 261-271.
2. A.G. Robling, F.M. Hinant, D.B. Burr, and C.H. Turner, *Shorter, more frequent mechanical loading sessions enhance bone mass*. *Medicine and science in sports and exercise*, 2002. **34**(2): p. 196-202.
3. A.M. de Leeuw, R. Graf, P.J. Lim, J. Zhang, G.N. Schädli, S. Peterhans, M. Rohrbach, C. Giunta, M. Rüger, and M. Rubert, *Physiological cell bioprinting density in human bone-derived cell-laden scaffolds enhances matrix mineralization rate and stiffness under dynamic loading*. *Frontiers in Bioengineering and Biotechnology*, 2024. **12**: p. 1310289.
4. G.N. Schädli, J.R. Vetsch, R.P. Baumann, A.M. de Leeuw, E. Wehrle, M. Rubert, and R. Müller, *Time-lapsed imaging of nanocomposite scaffolds reveals increased bone formation in dynamic compression bioreactors*. *Communications biology*, 2021. **4**(1): p. 110.
5. J. Zhang, J. Griesbach, M. Ganeyev, A.-K. Zehnder, P. Zeng, G.N. Schädli, A. de Leeuw, Y. Lai, M. Rubert, and R. Müller, *Long-term mechanical loading is required for the formation of 3D bioprinted functional osteocyte bone organoids*. *Biofabrication*, 2022. **14**(3): p. 035018.
6. C. Wittkowske, G.C. Reilly, D. Lacroix, and C.M. Perrault, *In vitro bone cell models: impact of fluid shear stress on bone formation*. *Frontiers in Bioengineering and Biotechnology*, 2016. **4**: p. 87.
7. M. Bernero, D. Zauchner, R. Müller, and X.-H. Qin, *Interpenetrating network hydrogels for studying the role of matrix viscoelasticity in 3D osteocyte morphogenesis*. *Biomaterials Science*, 2024.

Reviewers' Comments:

Reviewer #1:

Remarks to the Author:

The authors have thoroughly addressed the comments and further improved the quality of the manuscript. It is recommended being accepted for publication in Nature Communications.